# Toward neuroanatomical and cognitive foundations of macaque social tolerance grades

Sarah Silvere[1,2], Julien Lamy[3], Chrystelle Po[3], Mathieu Legrand[1,2], Jerome Sallet[4,5], Sebastien Ballesta[1,2]*

[1]Laboratoire de Neurosciences Cognitives et Adaptatives, UMR 7364, Strasbourg, France; [2]Centre de Primatologie de l'Université de Strasbourg, Niederhausbergen, France; [3]ICube (UMR 7357), Université de Strasbourg-CNRS, Strasbourg, France; [4]Univ Lyon, Université Lyon, Inserm, Stem Cell and Brain Research Institute, U1208, Bron, France; [5]Wellcome Center for Neuroimaging, Dpt of Experimental Psychology, University of Oxford, Oxford, United Kingdom

*For correspondence:
sebastien.ballesta@gmail.com

Competing interest: The authors declare that no competing interests exist.

## eLife Assessment

This **important** work compares the size of two brain areas, the amygdala and the hippocampus, across 12 species belonging to the Macaca genus. The authors find, using a **convincing** methodological approach, that amygdala - but not hippocampal - volume varies with social tolerance grade, with high tolerance species showing larger amygdala than low tolerance species of macaques. Interestingly, their findings also suggest an inverted developmental effect, with intolerant species showing an increase in amygdala volume across the lifespan, compared to tolerant species exhibiting the opposite trend. Overall, this paper offers new insights into the neural basis of social and emotional processing.

**Abstract** The macaque genus includes 25 species with diverse social systems, ranging from low to high social tolerance grades. Such interspecific behavioral variability provides a unique model to tackle the evolutionary foundation of primate social brain. Yet, the neuroanatomical correlates of these social tolerance grades remain unknown. To address this question, we expressed social tolerance grades within a novel cognitive framework and analyzed *post-mortem* structural scans from 12 macaque species. Our results show that amygdala volume is a subcortical predictor of macaques' social tolerance, with high tolerance species exhibiting larger amygdala than low tolerance ones. We further investigated the developmental trajectory of amygdala across social grades and found that intolerant species showed a gradual increase in relative amygdala volume across the lifespan. Unexpectedly, tolerant species exhibited a decrease in relative amygdala volume across the lifespan, contrasting with the age-related increase observed in intolerant species—a developmental pattern previously undescribed in primates. Taken together, these findings provide valuable insights into the cognitive, neuroanatomical, and evolutionary basis of primates' social behaviors.

## Introduction

A complex social environment implies a greater cognitive demand of social representations and interactions, which is one of the driving forces behind the evolution of the primate brain (*Dunbar, 2009*; *Freeberg et al., 2012*; *Heldstab et al., 2022*). Correlations between social environment and variations in brain structure volumes have been reported, both in humans (*Kanai et al., 2012*; *Maguire et al.,*

**eLife digest** Macaque monkeys live under a variety of social regimes. Some species flourish within highly structured, hierarchical societies, while others navigate more tolerant yet less predictable social networks. Primatologists have categorised these social differences, including how often reconciliation occurs after conflicts, into four levels of social tolerance. However, the neuronal mechanisms underlying these social variations remain poorly understood.

Closely related species offer a natural laboratory for studying the social brain in primates. To investigate how neural networks may have evolved in response to differing social challenges, Silvère et al. analysed 43 brain scans from 12 macaque species. All data were gathered from animals that had died of natural or accidental causes

The scans showed that the relative size of a species' amygdala – a brain region involved in emotional responses, decision-making, and memory – correlates with its level of social tolerance. For example, low-tolerance species are born with a smaller amygdala, which grows larger with age. Conversely, in more socially tolerant species, the amygdala decreases in size as they age, contrasting with findings in other primates, including humans.

These findings imply that living in a more tolerant social environment could impose greater cognitive demands on the brain, with the amygdala possibly playing a part in complex social cognition. In contrast, the volume of a brain region called the hippocampus revealed more variable differences across social grades among macaques, with a more significant effect observed only in individuals aged between 13 and 18 years. Additionally, differences in hippocampal volume also varied among monkeys living in different areas, supporting the idea that certain regions contribute to social cognitive processes in tolerant species, particularly during developmental phases linked to social maturation.

Exploring natural variation in brain evolution and function opens new avenues for primate neuroscience. A more extensive comparative analysis across all living primate species could further clarify evolutionary pathways. Moreover, identifying neural networks that are either evolutionarily conserved or highly variable may help shape new research directions aimed at understanding the biological basis of neurodivergence.

*2000*; *Parkinson et al., 2017*) and in non-human primates (NHP; *Noonan et al., 2014*; *Sallet et al., 2011*; *Meguerditchian et al., 2021*; *Testard et al., 2022*). In rhesus macaques (*Macaca mulatta*), previous studies have demonstrated that interindividual variation in social characteristics—such as hierarchical status (*Noonan et al., 2014*) or group size (*Sallet et al., 2011*; *Testard et al., 2022*) – is associated with grey matter volume in core regions of the social brain, including the amygdala, the hippocampus, the superior temporal sulcus (STS), and the rostral prefrontal cortex (rPFC). Supporting the broader relevance of these findings across *Cercopithecinae*, a study in olive baboons (*Papio anubis*) revealed that individuals living in larger social groups exhibited greater total brain volumes, with an effect primarily driven by white matter (*Meguerditchian et al., 2021*).

Despite the existence of 25 species within the *Macaca* genus (*Cooper et al., 2022*; *Cords, 2012*; *Ghosh et al., 2022*; *Thierry, 2007*; *Thierry et al., 2004*), most neuroscience research focuses on two species, *M. mulatta* and *Macaca fascicularis* (and in rare cases *Macaca nemestrina* and *Macaca fuscata Carlo et al., 2010*; *Isa et al., 2009*; *Maranesi et al., 2014*). In spite of the relatively short evolutionary divergence time within this genus (6–8 million years *Perelman et al., 2011*), the various macaque species display a considerable interspecific variety of social behaviors while usually maintaining a multi-male, multi-female, and multi-generational social structure (*Balasubramaniam et al., 2018*; *Thierry, 2007*; *Thierry and Sapolsky, 2000*). These behavioral differences are characterized by different styles of dominance (*Balasubramaniam et al., 2012*), severity of agonistic interactions (*Duboscq et al., 2014*), nepotism (*Thierry and Berman, 2010*; *Duboscq et al., 2013*; *Sueur et al., 2011*), and submission signals (*de Waal and Luttrell, 1985*; *Rincon et al., 2023*), among the 18 covariant behavioral traits described in Thierry's classification of social tolerance (*Thierry, 2021*; *Thierry, 2017*; *Thierry and Sapolsky, 2000*).

Despite this large behavioral variability, macaque species display broadly similar general cognitive abilities (*Aguenounon et al., 2022*). Specific differences observed in domains such as inhibitory control or social flexibility are thus more likely to reflect adaptive responses to species-specific social

constraints, rather than intrinsic disparities in overall intelligence (*Joly et al., 2017*; *Loyant et al., 2023*). Altogether, the socio-behavioral diversity within the *Macaca* genus provides a compelling model to investigate how social ecology shapes cognition and its neural substrates.

The concept of social tolerance, central to this comparative approach, has sometimes been used in a vague or unidimensional way. As *Thierry, 2021* pointed out, the notion was initially constructed around variations in agonistic relationships – dominance, aggressiveness, appeasement, or reconciliation behaviors – before being expanded to include affiliative behaviors, allomaternal care, or male–male interactions (*Thierry, 2021*). These traits do not necessarily align along a single hierarchical axis but rather reflect a multidimensional complexity of social style, in which each trait may have co-evolved with others (*Thierry, 2021*; *Thierry and Sapolsky, 2000*; *Thierry et al., 2004*). Moreover,

**Table 1.** Cognitive and neuroanatomical categorization of behavioral traits associated with macaque social tolerance.

| Social trait | Underlying social consequences | Cognitive dimension | Neural correlate |
|---|---|---|---|
| Complexity of communication system | Demands in interpreting social signals and adjusting communication to context (*Liebal et al., 2014*). | Higher socio-cognitive demands | Amygdala volume higher (*Bickart et al., 2011*; *Sallet et al., 2011*); Hippocampus volume higher (*Kanai et al., 2012*; *Todorov et al., 2019*)[9,10] |
| Rate of reconciliation | Demands in recalling social history and regulating affiliation (*Thierry and Sapolsky, 2000*). | | |
| Male-to-male coalitions | Demands in social knowledge and strategic social decisions (*Petit et al., 1997*; *Silk, 1999*). | | |
| Cooperative behaviors | Demands in understanding intentions and coordinating actions during interactions (*Demaria and Thierry, 2001*; *Micheletta et al., 2012*). | | |
| Intensity of aggression | Demands in inhibiting impulsive behaviors and regulating emotions (*Adams et al., 2015*; *Loyant et al., 2023*). | Better inhibitory control | Amygdala volume lower (*Tottenham et al., 2010*); Hippocampus volume unchanged (*Tottenham et al., 2010*). |
| Confidence of social play | Demands in adjusting behavior and inhibiting responses in mutual interactions (*Petit et al., 2008*; *Scopa and Palagi, 2016*). | | |
| Resource distribution evenness | Demands in adjusting behavior during competitive interactions and regulating emotions (*Thierry, 2007*). | | |
| Kin bias (nepotism) | Kin knowledge is less informative to predict social relationships (*Silk, 2002*; *Silk et al., 2003*). | Lower predictability of social environment (heightened chronic stress) | Amygdala volume higher (*Bickart et al., 2014*; *Sallet et al., 2011*; *Tottenham et al., 2010*); Hippocampus volume lower (*Kim et al., 2015*; *Lyons et al., 2001*; *Meyer and Hamel, 2014*). |
| Dominance asymmetry | Conflicts are not always won by dominants, leading to greater outcome unpredictability (*de Vries et al., 2006*; *Thierry, 2007*). | | |
| Formal submission signals | Communication during conflict is less predictive of outcomes (*Flack et al., 2006*; *Waller et al., 2013*). | | |
| Intensity of female rank inheritance | Matrilinear knowledge is less informative to predict social relationships (*Hill and Okayasu, 1995*; *Kutsukake, 2000*). | | |
| Rate of affiliative contact | Affiliative networks are denser, reducing predictability (*Duboscq et al., 2013*; *Massen et al., 2010*; *Silk et al., 2003*). | | |
| Rate of counter-aggression | Subordinates are more likely to retaliate, making social outcomes less predictable (*Balasubramaniam et al., 2012*; *Petit et al., 1997*). | | |
| Rate of immature interference in mating | Mounting behaviour increases social interactions, producing more erratic social patterns (*Petit et al., 2008*). | | |
| Centrality of top-ranking males | Low centrality of top-ranking males decreased social network predictability (*Sueur et al., 2011*). | | |
| Mother protectiveness | Limits how much infants interact with other group members (*Maestripieri, 1994*). | Unclassified | / |
| Allomothering behavior | Reciprocal benefits for females and infants (*Fairbanks, 1990*). | | |
| Delayed male dispersal | Limits the range of social networks open to individuals (*Thierry, 2007*). | | |

the lack of a standardized scientific definition has sometimes led to labeling species as 'tolerant' or 'intolerant' without explicit criteria (*Gumert and Ho, 2008*; *Patzelt et al., 2014*).

To ground the investigation of social tolerance in a comparative neuroanatomical framework, we introduced a tentative working model that articulates behavioral traits, cognitive dimensions, and their potential subcortical neural substrates. Drawing upon 18 behavioral traits identified in Thierry's comparative analyses (*Thierry, 2021*; *Thierry, 2007*), we organized these traits into three core dimensions: socio-cognitive demands, behavioral inhibition, and the predictability of the social environment (*Table 1*). This conceptualization did not aim to redefine social tolerance itself, but rather to provide a structured basis for testing neuroanatomical hypotheses related to the volume of relevant subcortical areas and social style variability. It echoes recent efforts to bridge behavioral ecology and cognitive neuroscience by linking specific mental abilities – such as executive functions or metacognition – with distinct prefrontal regions shaped by social and ecological pressures (*Bouret et al., 2024*; *Testard et al., 2022*).

Navigating social life in primate societies requires substantial cognitive resources: individuals must not only track multiple relationships, but also regulate their own behavior, anticipate others' reactions, and adapt flexibly to changing social contexts. Taking advantage of databases of magnetic resonance imaging (MRI) structural scans, we conducted the first comparative study integrating neuroanatomical data and social behavioral data from closely related primate species of the same genus to address the following questions: To what extent can differences in volumes of subcortical brain structures be correlated with varying degrees of social tolerance? Additionally, we explored whether these dispositions reflect primarily innate features, shaped by evolutionary processes, or acquired through socialization within more or less tolerant social environments.

The first category, socio-cognitive demands, refers to the cognitive resources needed to process, monitor, and flexibly adapt to complex social environments. Linking those parameters to neurological data is at the core of the social brain theory (*Dunbar, 2009*). Macaques' social systems require advanced abilities in social memory, perspective-taking, and partner evaluation (*Freeberg et al., 2012*). This is particularly true in tolerant species, where the increased frequency and diversity of interactions may amplify the demands on cognitive tracking and flexibility. Tolerant macaque species typically live in larger groups with high interaction frequencies, low nepotism, and a wider range of affiliative and cooperative behaviors, including reconciliation, coalition-building, and signal flexibility (*Thierry, 2021*; *Thierry and Sapolsky, 2000*). Tolerant macaque species also exhibit a more diverse and flexible vocal and facial repertoire than intolerant ones, which may help reduce ambiguity and facilitate coordination in dense social networks (*Rincon et al., 2023*; *Scopa and Palagi, 2016*; *Rebout et al., 2020*). Experimental studies further show that macaques can use facial expressions to anticipate the likely outcomes of social interactions, suggesting a predictive function of facial signals in managing uncertainty (*Micheletta et al., 2012*; *Waller et al., 2016*). Even within less tolerant species, like *M. mulatta*, individual variation in facial expressivity has been linked to increased centrality in social networks and greater group cohesion, pointing to the adaptive value of expressive signaling across social styles (*Whitehouse et al., 2024*).

The second category, inhibitory control, includes traits that involve regulating impulsivity, aggression, or inappropriate responses during social interactions. Tolerant macaques have been shown to perform better in tasks requiring behavioral inhibition and also express lower aggression and emotional reactivity than intolerant macaques both in experimental and in natural contexts (*Joly et al., 2017*; *Loyant et al., 2023*). These features point to stronger self-regulation capacities in species with egalitarian or less rigid hierarchies. More broadly, inhibition – especially in its strategic form (self-control) – has been proposed to play a key role in the cohesion of stable social groups. Comparative analyses across mammals suggest that this capacity has evolved primarily in anthropoid primates, where social bonds require individuals to suppress immediate impulses in favor of longer-term group stability (*Dunbar and Shultz, 2025*). This view echoes the conjecture of *Passingham et al., 2012*, who proposed that the expansion of lateral prefrontal area BA10 in anthropoids enabled the kind of behavioral flexibility needed to navigate complex social environments (*Passingham et al., 2012*).

The third category, social environment predictability, reflects how structured and foreseeable social interactions are within a given society. In tolerant species, social interactions are more fluid and less kin-biased, leading to greater contextual variation and role flexibility, which likely imply a sustained level of social awareness. In fact, as suggested by recent research, such social uncertainty and prolonged

incentives are reflected by stress-related physiology: tolerant macaques such as *M. tonkeana* display higher basal cortisol levels, which may be indicative of a chronic mobilization of attentional and regulatory resources to navigate less predictable social environments (**Sadoughi et al., 2021**).

Each behavioral trait was individually evaluated based on existing empirical literature regarding the types of cognitive operations it likely involves. When a primary cognitive dimension could be identified, the trait was assigned accordingly. However, some behaviors – such as maternal protection, allomaternal care, or delayed male dispersal – do not map neatly onto a single cognitive process. These traits likely emerge from complex configurations of affective and social-motivational systems and may be better understood through frameworks such as attachment theory (**Suomi, 2008**), which emphasizes the integration of social bonding, emotional regulation, and contextual plasticity. While these dimensions fall beyond the scope of the present framework, they offer promising directions for future research, particularly in relation to the hypothalamic and limbic substrates of social and reproductive behavior.

Rather than forcing these traits into potentially misleading categories, we chose to leave them unclassified within our current cognitive framework. This decision reflects both a commitment to conceptual clarity and the recognition that some behaviors emerge from a convergence of cognitive demands that cannot be neatly isolated. This tripartite framework, leaving aside reproductive-related traits, provides a structured lens through which to link behavioral diversity to specific cognitive processes and generate neuroanatomical predictions.

We therefore associated these three categories with neuroanatomical hypotheses regarding variations in the volume of two subcortical structures of interest, the amygdala and the hippocampus. Based on existing literature on the effects of the socio-cognitive demands, inhibitory control, and social unpredictability – particularly when it induces sustained activation of stress-related systems – on the volume of these two regions (**Bickart et al., 2011**; **Caetano et al., 2021**; **Coplan et al., 2014**; **Haley et al., 2012**; **Howell et al., 2014**; **Lyons et al., 2001**; **Sallet et al., 2011**), we hypothesized that increased cognitive demands from the social environment could lead to a differential effect on amygdala and hippocampus volumes (**Bickart et al., 2011**; **Haley et al., 2012**; **Lyons et al., 2001**; **Sallet et al., 2011**). We summarized our working hypotheses into a table comparing grade 4 and grade 1 species (**Table 1**).

This table summarizes the 18 behavioral traits used to characterize social tolerance grades in the *Macaca* genus, based on Thierry's comparative framework. Each trait is associated with a description of its underlying cognitive implications and assigned—when applicable—to one of three cognitive dimensions: (i) socio-cognitive demands (e.g. tracking partners, coordinating actions), (ii) behavioral inhibition (e.g. regulating impulsivity), or (iii) predictability of the social environment (e.g. anticipating interaction outcomes). The final column presents the hypothesized effects of these dimensions on the volume of two subcortical structures: the amygdala and the hippocampus. Traits that could not be clearly assigned to a specific cognitive dimension—often related to maternal or reproductive strategies—are marked as 'unclassified'. This framework is used to generate testable predictions about the neural substrates of social style diversity in macaques.

We tested our hypothesis using 42 *post-mortem* MRI acquisitions of 12 macaque species representing the four grades of social tolerance. The dataset was both composed of samples from open access databases (**Milham et al., 2018**; **Navarrete et al., 2018**; **Sakai et al., 2018**) as well as newly and unpublished samples from the collection of the Centre de Primatologie de l'Université de Strasbourg (CdP) and INSERM-Oxford University. These samples include brain images of *Macaca tonkeana* and *Macaca thibetana,* two macaque species that have never been scanned before as well as a scan of *Macaca nigra* that is rare in the existing literature (**Navarrete et al., 2018**; **Sakai et al., 2023**). Up to this date, only one study has included tolerant species of macaque monkeys in such neuroanatomical comparative framework (**Jones et al., 2021**). While **Jones et al., 2021** identified interspecific differences in amygdala microstructure and serotonergic innervation, their histological approach did not assess structural volumes at the whole-brain level. To our knowledge, our study is the first to report neuroanatomical correlates of social tolerance grades of the *Macaca* genus based on *post-mortem* MRI volumetric analysis. This approach reveals two key findings. First, across species, amygdala volume is positively correlated with social tolerance grades, with more tolerant macaque species exhibiting larger amygdala volumes. Second, developmental trajectories of the amygdala diverge according to social style: in intolerant species, amygdala volume increases with age – as commonly reported

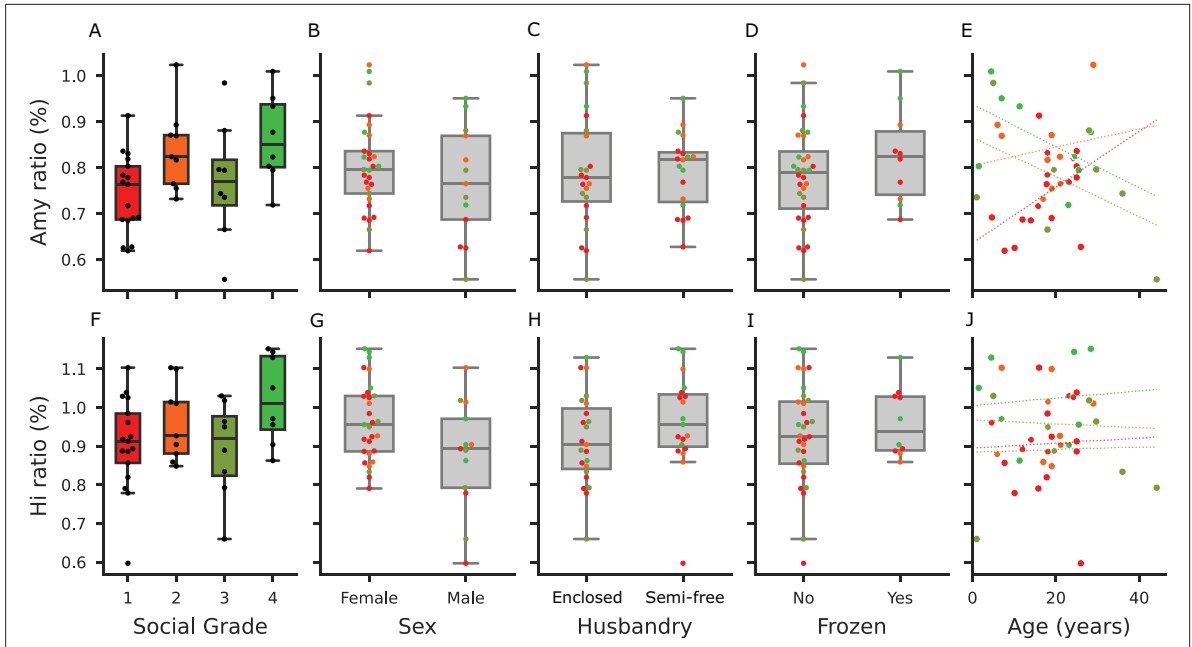

**Figure 1.** Model predictors of the amygdala and hippocampus, and volume predictions across social tolerance grades. First row (**A**–**D**): Model predictors and responses for amygdala volume. The volume ratio is calculated as the amygdala volume divided by the total brain volume (excluding the myelencephalon and cerebellum). (**A**) Distribution of amygdala volume ratios across social tolerance grades. (**B**) Distribution of amygdala volume ratios by sex. (**C**) Distribution of amygdala volume ratios by husbandry condition (enclosed vs. semi-free). (**D**) Distribution of amygdala volume ratios by the frozen status. (**E**) Distribution of amygdala volume ratios by age. Second row (**F**–**J**): Model predictors and responses for hippocampal volume. The volume ratio is calculated as the hippocampal volume divided by the total brain volume (excluding the myelencephalon and cerebellum). (**F**) Distribution of hippocampal volume ratios across social tolerance grades. (**G**) Distribution of hippocampal volume ratios by sex. (**H**) Distribution of hippocampal volume ratios by husbandry condition (enclosed vs. semi-free). (**I**) Distribution of hippocampal volume ratios by the frozen status. (**I**) Distribution of hippocampal volume ratios by age. Panels **A**-**E** and **F**-**J** share the same y-axis.

The online version of this article includes the following figure supplement(s) for figure 1:

**Figure supplement 1.** Total brain volume across macaque species categorized by social grade.

**Figure supplement 2.** Sequence of dissection steps and MRI acquisition.

**Figure supplement 3.** Set of photographs of the preparation of the fixed brain for MRI acquisitions.

in the literature (*Schumann et al., 2019*) – whereas in tolerant species, we observe an unexpected marked decrease over the lifespan. This study offers a novel and valuable perspective by comparing interspecies brain structures to investigate the functioning of the social brain, while accounting for key socio-cognitive variables.

## Results

We obtained structural MRI scans of 42 macaques of 12 macaque species. Using a semi-automated registration to an atlas (SARM, *Hartig et al., 2021*), we extracted amygdala and hippocampus volumes and analyzed whether these covaried with social grade and age, using a Bayesian model. The raw relations between the main response variables (the amygdala's and hippocampus volumes) are depicted in *Figure 1*.

### Model quality and coefficients

The R² coefficient of determination of the model indicated a large proportion of variability accounted for by the model (90% credible interval: [0.87, 0.97]). The effect of sex was minimal for the amygdala but more pronounced for the hippocampus (*Figures 1 and 2*), whereas husbandry had a limited effect on both regions of interest. Amygdala volume increased with social grade (independently of its interaction with age) and with age (independently of its interaction with social grade). However, the interaction between social grade and age suggested that the trajectory of amygdala volume over the

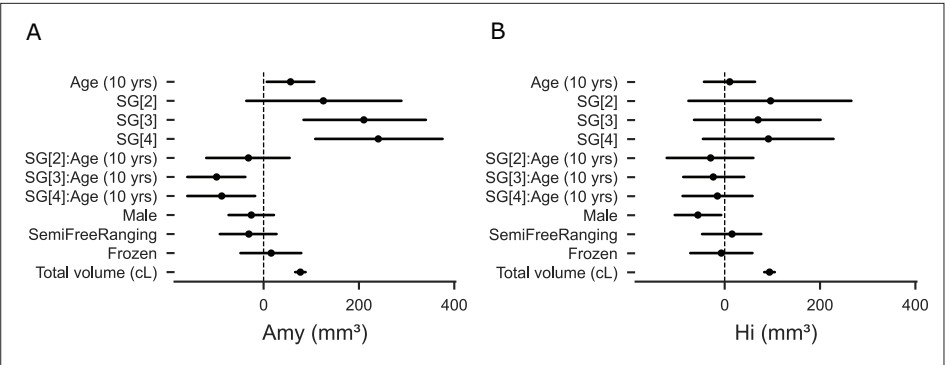

**Figure 2.** Parameters of the model. (**A**) Parameters of the model for the amygdala volume. (**B**) Parameters of the model for the hippocampal volume. SG [x]: Social Grade [x] vs Social Grade [1]; SG[x]: Age (10 years): Social Grade-Age interaction.

lifespan differs across social grades, as detailed below. Total brain volume was included as a covariate in the model to account for interindividual differences in brain size. For descriptive purposes, its distribution across social grades is shown in *Figure 1—figure supplement 1*.

Despite limited sample size, the interaction between social grade and age suggested a differential trajectory of amygdala volume across the lifespan among different social grades (*Figures 1 and 2*).

To further assess group differences, we implemented Bayesian hypothesis testing using a Region of Practical Equivalence (ROPE, *Kruschke, 2015*) approach, with the ROPE defined as ± 0.1 × σ. This method allows classification of results into three categories: (a) a credible difference if the entire posterior interval lies outside the ROPE; (b) an absence of difference if it lies entirely within the ROPE; and (c) inconclusive if it overlaps the ROPE. For the amygdala, social grade 4 (SG4, i.e. tolerant) individuals had credibly larger volumes than social grade 1 (SG1, i.e. intolerant) individuals up to 19 years of age. For the hippocampus, the posterior distribution of the SG4–SG1 difference briefly exceeded the ROPE between approximately 13 and 18 years of age, indicating a credible difference in this age

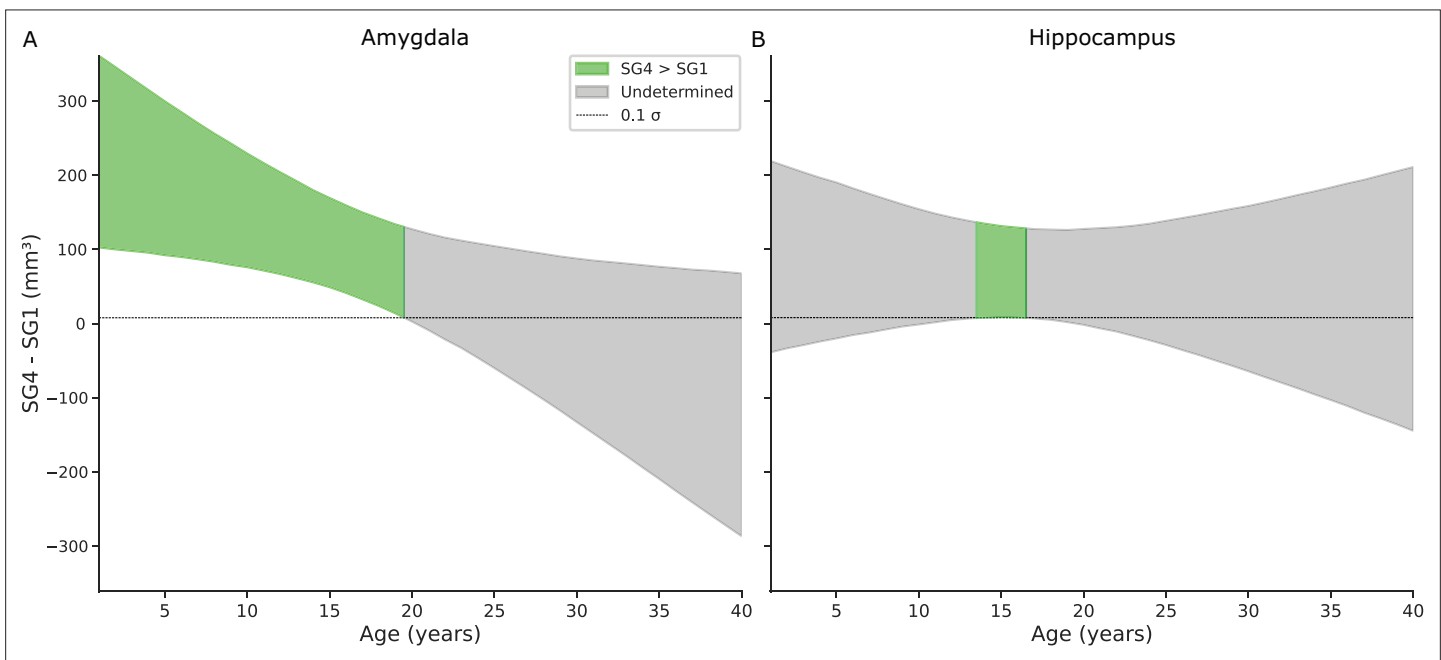

**Figure 3.** Bayesian hypothesis testing using a Region of Practical Equivalence (ROPE) to assess volume (in mm³) differences between Social Grade 4 (SG4; tolerant) and Social Grade 1 (SG1; intolerant) across age, for the amygdala (left) and hippocampus (right). Curves represent median posterior estimates, and shaded areas show 90% credible intervals. Gray bands indicate the ROPE (±0.1σ). For the amygdala, the difference is credible until ~19 years. For the hippocampus, a credible effect is observed only between ~13 and 18 years.

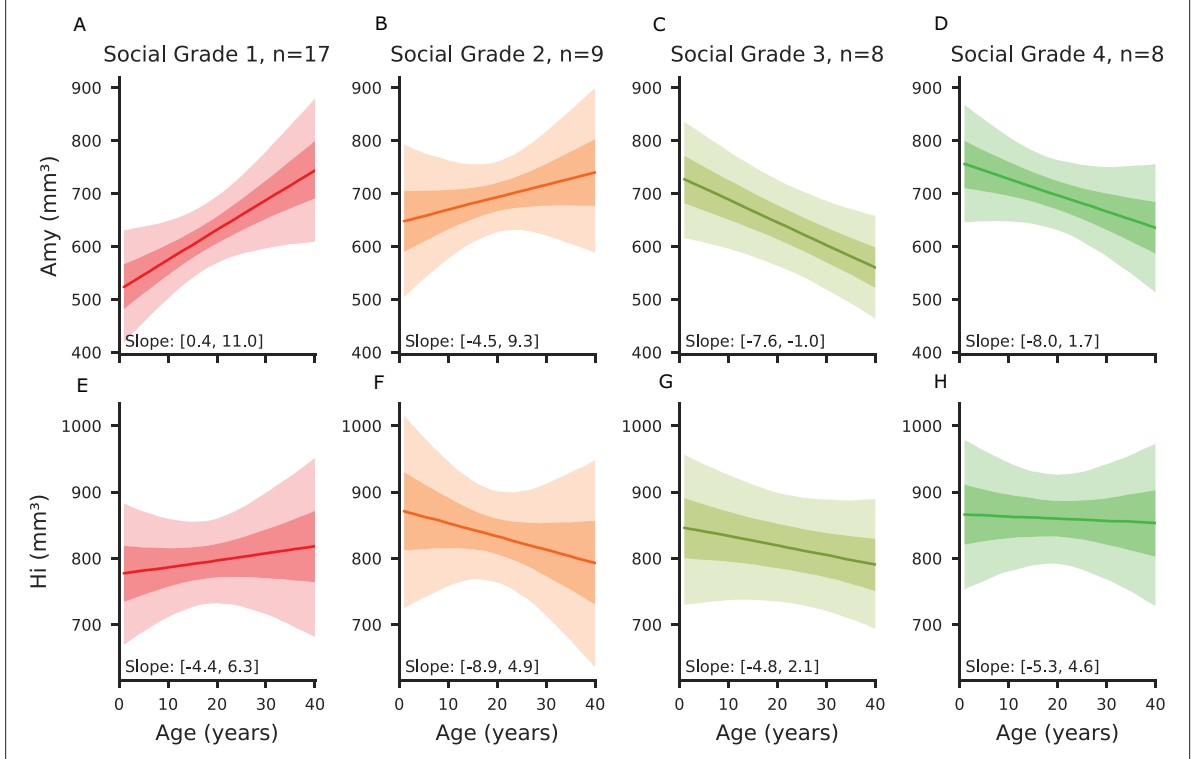

**Figure 4.** Volume predictions across social tolerance grades of the amygdala and hippocampus. All panels represent the predictions of the multivariate Bayesian linear model, where all the variables are kept constant (including total brain volumes) in order to represent the effect of age only on the volume of amygdala and hippocampus in mm³. First row (**A–D**): Predicted amygdala volume across social tolerance grades over the lifespan. (**A**) Predicted amygdala volume as a function of age for grade 1 (intolerant) individuals. (**B**) Predicted amygdala volume as a function of age for grade 2 individuals. (**C**) Predicted amygdala volume as a function of age for grade 3 individuals. (**D**) Predicted amygdala volume as a function of age for grade 4 (tolerant) individuals. Second row (**E–H**): Predicted hippocampal volume across social tolerance grades over the lifespan. (**E**) Predicted hippocampal volume as a function of age for grade 1 individuals. (**F**) Predicted hippocampal volume as a function of age for grade 2 individuals. (**G**) Predicted hippocampal volume as a function of age for grade 3 individuals. (**H**) Predicted hippocampal volume as a function of age for grade 4 individuals. In the plots, the solid lines represent the mean predicted values, and the shaded areas indicate the 90% credible intervals, with each social grade shown in a distinct color.

window. Outside this range, the intervals overlapped the ROPE, resulting in inconclusive evidence. However, the 90% posterior intervals remained entirely above zero at all ages, indicating that SG1 individuals never had larger hippocampal volumes than SG4 (*Figure 3*).

## Predicted data

*Figure 4* illustrates how amygdala volume development varies with an individual's social grade over their lifespan. Two results stand out: first, individuals in Social Grade 1 showed a distinct pattern of amygdala volume development compared to other social grades. Although Grade 1 individuals had a smaller amygdala volume in early years compared to the other Social Grades, the amygdala's volume variation slope was steeper than for the other Grades (slope with 90% credible intervals [0.6, 11.0]). This increase contrasted with trends observed in Grades 3 (slope with 90% CI [-7.6,–0.9]) and 4 (slope with 90% CI [–8.0, 1.9]), which showed a decrease in volume over time. Individuals in Grade 2 also showed a slight increase in amygdala volume (slope with 90% CI [–4.4, 9.3]), similar to grade 1 but not as steep.

When comparing Grade 1 and Grade 4, individuals in Grade 4 showed larger amygdala volumes until approximately 19 years of age (*Figure 4*).

As expected from model predictions, hippocampal volume showed limited variation across age and social grades. ROPE-based hypothesis testing revealed that hippocampal volume in SG1 individuals was never greater than in SG4 individuals, supporting a consistent asymmetry in favor of more tolerant species.

## Discussion

We studied for the first time the neuroanatomical foundation of the naturally observed diversity of behavioral traits within the *Macaca* genus. We have assembled a unique database representing nearly half of the known macaque species, with a variety of ages, sexes, and origins. 12 species of them had never been scanned prior to our study. Our investigation focused on the subcortical structures of the brain and more especially the amygdala. This set of nuclei, sometimes referred to as a hub of brain networks related to sociality and their social lives, is well known for their roles in the stress response (*Hölzel et al., 2010*), emotional regulation, and social cognition (*Noonan et al., 2014*; *Bickart et al., 2014*; *Amaral et al., 2003*). Based on *post-mortem* MRI acquisitions from 12 of the 25 macaque species, we showed that amygdala volume correlated with the social tolerance grade and increased with the level of the social grade. Secondly, grade 4 species had a significantly higher amygdala volume at the start of their lives, which decreased over time, compared with grade 1 species, which showed the opposite trend. Finally, further hypothesis testing suggested that species of grade 1 never exhibited larger hippocampus and may have smaller hippocampus around age 15 when compared to those of grade 4. In accordance with our hypotheses (*Table 1*), our findings substantiated the assertions that (i) social tolerance is rooted in neuroanatomical differences that can be detected at an early stage of individuals' development, (ii) social styles exert differential influence on subcortical structures throughout individuals' lifespan and (iii) such phenomena should be mainly driven by the socio-cognitive demands that vary with species social style (as evidenced by the higher amygdala and hippocampus volumes in higher tolerance species).

### A neuroanatomical account for social tolerance differences

The social tolerance grades are based on previous ethological observations of behaviors across different species of the genus (*Thierry, 2021*; *Thierry, 2007*). From these observations, we identified three major cognitive processes—socio-cognitive demands, social predictability, and inhibitory control (see *Table 1*)—that underpin the observed behaviors. Among the behaviors we classified with 'high social-cognitive demand', several have been previously described in the literature as particularly discriminating between grades 1 and 4. It included greater social network density in grade 4 species (a consequence of, *inter alia*, low nepotism in tolerant species, facilitating interactions between unaffiliated conspecifics) (*Balasubramaniam et al., 2018*; *Sueur et al., 2011*), more complex facial mimics as well as a more complex communication system (*Dobson, 2012*; *Rincon et al., 2023*; *Scopa and Palagi, 2016*; *Zannella et al., 2017*), a significantly higher rate of reconciliation (*Thierry, 2007*), and a higher frequency of cooperative behaviors, including male-to-male coalition behaviors (*De Waal and Luttrell, 1989*; *Thierry et al., 2008*) in grade 4 species.

At first glance, one may presume that species of lower social tolerance level, that displayed more overall aggressive behaviors, would have a larger amygdala when compared to more tolerant species. In fact, amygdala ablation or activity modulation in macaque monkeys showed that animals displayed less aggressive and different patterns of social behaviors (*Amaral, 2002*; *Bliss-Moreau et al., 2017*; *Wellman et al., 2016*; *Forcelli et al., 2017*; *Raper et al., 2017*), supporting the idea that amygdala activity can promote aggressive behaviors. Our study revealed an opposite trend; amygdala was found to be larger in more tolerant species, and this apparent contradiction invites a more integrative view of the amygdala (*Adolphs, 2009*; *Amaral et al., 2003*; *Pessoa, 2010*), not only as a relay for emotional reactivity, but as a multifunctional hub embedded in complex social networks (*Bickart et al., 2014*). Such complex patterns of behavioral implications are also reflected at the cellular level, as the amygdala is composed of several different nuclei that are broadly connected with other brain areas that may display opposite functions (*Zikopoulos et al., 2016*; *Amiez et al., 2023*). Rather than opposing social cognition and emotion, our results support the view that emotional processing is deeply intertwined with social function—both being subserved by overlapping neural circuits (*Domínguez-Borràs and Vuilleumier, 2022*). It also suggests that additional neural mechanisms, particularly those involving prefrontal and anterior cingulate regions implicated in the top-down regulation of affect and social behavior, may contribute to shaping species differences in social tolerance (*Ochsner and Gross, 2008*). While our analysis compares social tolerance grades with variations in brain structure, the originality of our framework also lies in introducing three cognitive dimensions that bridge behavioral traits and neural substrates. This intermediate level of interpretation allows us

to move beyond simple grade-to-structure associations, toward a more mechanistic understanding of the links between social behavior, cognition, and neuroanatomy.

Amygdala volume has also been shown to correlate positively with social network complexity in grade 1 species, as measured by the social network size of individuals (*Sallet et al., 2011*; *Testard et al., 2022*), or by the social status of the animals (*Noonan et al., 2014*). This supports the idea that the amygdala is sensitive to both structural social features and dynamic aspects of social networks.

## Developmental trajectories and life-history plasticity

We are then led to question the origin of the social tolerance effect on amygdala volume, not in terms of a rigid nature versus nurture dichotomy, but in terms of differential developmental trajectories. Cross-fostering experiments (*de Waal and Johanowicz, 1993*), along with our own results, suggest that social tolerance grades reflect both early, possibly innate predispositions and later environmental shaping. Moreover, the behavioral shifts observed in cross-fostered individuals underscore the plasticity of social style acquisition and the role of early social environment in shaping neural substrates of social behavior. Notably, tolerant species exhibit larger amygdala volumes early in life, while intolerant species show a progressive increase across the lifespan—a pattern that suggests a dual influence of biological programming and cumulative social experience. These environmental influences likely arise from both species-specific social dynamics—such as variations in affiliative behavior and social play (*Beltrán Francés et al., 2020*)—and broader ecological conditions that structure the demands of social life. The age-related volumetric changes we observed, particularly the divergence in developmental trajectories between tolerant and intolerant species, reinforce this idea and echo previous reports of amygdala growth patterns in humans and macaques (*Schumann et al., 2019*; *Uematsu et al., 2012*). Taken together, these elements support the view that social tolerance is not fixed but emerges from the interplay between inherited developmental programs and the specific socio-ecological environments in which individuals mature.

Notably, the developmental trajectory of the amygdala in tolerant species does not align with that of intolerant species or with human developmental patterns (*Schumann et al., 2019*; *Uematsu et al., 2012*). This finding suggests that neurodevelopmental pathways may exhibit significant variation among phylogenetically closely related primate species, potentially serving as an effective evolutionary target for adapting socio-ecological behaviors to environmental demands. Moreover, we observed that in old individuals (typically above 19 years), relative amygdala volume in grade 1 species could match that of grade 4 species — despite being significantly smaller earlier in life. Due to a limited sample size of our study, this crossing trend, already accounted for by our continuous age model, should be further investigated. These results call for cautious interpretation of age-related variation and further emphasize the importance of longitudinal studies integrating both behavioral, cognitive, and anatomical data in non-human primates, which would help to better understand the link between social environment and brain development (*Song et al., 2021*).

## Hippocampal volume and social cognitive demands in tolerant species

A credible difference in hippocampal volume favoring SG4 individuals was only revealed between approximately 13 and 18 years of age by our hypothesis testing using a ROPE framework. Outside this range, the difference remained overall positive but inconclusive. This restricted window of significance, along with the unidirectional trend across the lifespan, suggests that increased hippocampal volume may nonetheless be associated with higher social tolerance, at least in adulthood. At first glance, this observation may appear to contrast with literature linking chronic stress to reduced hippocampal size (*Kim et al., 2015*; *Lyons et al., 2007*). However, as previously discussed, *M. tonkeana* (a high-tolerance species) combines elevated basal cortisol levels with a relatively large hippocampus (*Sadoughi et al., 2021*; *Vandeleest et al., 2016*), which suggests that glucocorticoid exposure alone does not account for hippocampal variation in this context. Instead, our findings are more consistent with the idea that hippocampal structure reflects species-specific cognitive demands associated with navigating complex and tolerant social environments—such as spatial memory, social recognition, or contextual learning (*Han et al., 2021*; *Sallet et al., 2011*). Within the conceptual framework introduced in this study, these results point to the importance of socio-cognitive requirements—rather than social environmental unpredictability or behavioral inhibition abilities—as potential drivers of interspecific variation in hippocampal anatomy. Comparative measurements and observations at the

individual level along with in vivo MRI from these same considered individuals may help to further understand how social tolerance can relate to cognitive abilities and its neural underpinning.

## Limits of the study and future directions

While our dataset is comprehensive in terms of the number of macaque species included, certain limitations must be acknowledged. For instance, phylogenetic analyses were beyond the reach of this study and integrating these statistical approaches could clarify the extent to which interspecific differences in brain structure and social behavior are due to shared ancestry or convergent evolution (*Ghosh et al., 2022*; *Heuer et al., 2025*).

Although we explained some interspecies variability, adding subjects to our database will increase statistical power and will help address potential confounding factors such as age or sex in future studies. One will benefit from additional information about each subject. While considered in our modeling, the social living and husbandry conditions of the individuals in our dataset remain poorly documented. The living environment has been considered, and the size of social groups for certain individuals, particularly for individuals from the CdP, has been recorded. However, these social characteristics have not been determined for all individuals in the dataset. As previously stated, the social environment has a significant impact on the volumetry of certain regions. Furthermore, there is a lack of data regarding the hierarchy of the subjects under study and the stress they experience in accordance with their hierarchical rank and predictability of social outcomes position (*McCowan et al., 2022*). In addition, our treatment of sex differences was limited. Although sex was included as a covariate in the Bayesian models, the strong imbalance in our dataset—favoring females (2:1 ratio)—precludes robust conclusions about sex-specific trajectories. Some trends, particularly regarding hippocampal volume, suggest potential interactions between sex, age, and social grade, but these effects remain exploratory. Addressing them adequately would require larger and more balanced samples, along with behavioral or hormonal data to capture intra-sexual variability. It is therefore important to recognize that confirmation of our findings should be achieved by analyzing datasets in which all of these confounding factors can be controlled more effectively.

While our study identifies the amygdala as a key subcortical structure associated with interspecific variation in social tolerance, it is important to acknowledge several neuroanatomical limitations. First, our analyses were conducted on the amygdala as a whole, without distinction between its internal nuclei. Although we used the SARM atlas (*Hartig et al., 2021*), which offers a high-quality parcellation for *M. mulatta*, the precision of this template does not allow for fully reliable automatic segmentation of amygdala subnuclei across the diverse species included in our dataset. As a result, our volumetric measures may conflate distinct functional subregions, potentially masking more localized effects. In this context, histological approaches remain essential for characterizing fine-grained neuroanatomical differences, as illustrated by *Jones et al., 2021*, who reported interspecific variation in cell density and serotonergic innervation within the amygdala (*Jones et al., 2021*). Future studies combining MRI-based volumetry with *post-mortem* histology would allow more precise identification of which subregions underlie the observed differences in social tolerance.

## Cognitive and neural perspectives on our understanding of social tolerance

Future directions linking behavior, cognition, and neuroanatomy could deepen our understanding of the roots of social tolerance among macaque species. This could lead to a better operationalization of the concept that could be applied to a wider range of non-human primate species. From a neural perspective, studying the cortical regions associated with social tolerance represents a promising yet ambitious goal. In fact, there is a variability within primate species in cerebral organization (*Amiez et al., 2023*; *Amiez et al., 2019*), which is likely to be found across the *Macaca* genus. Considering this cerebral variability would require extensive efforts to properly assess interspecies differences, making it beyond the scope of the current study that focuses on subcortical areas. However, as a starting point, exploring the connections between the amygdala, hippocampus, and medial prefrontal cortex could provide crucial insights into the neural correlates of social tolerance. These regions are central to stress regulation, socio-cognitive processing, and decision-making, all of which are likely impacted by social tolerance grades (*Caetano et al., 2021*; *Coplan et al., 2014*; *Kim et al., 2015*; *Phelps and LeDoux, 2005*; *Sapolsky et al., 1990*). In humans, repeated positive or stressful experiences have

been demonstrated to alter the size of subcortical brain areas such as the hippocampus or amygdala (*Davidson and McEwen, 2012*) and impair neuroplasticity (*Phelps, 2006*). Neuronal plasticity and learning have been identified as contributing factors to variations in the ROI volume, including the amygdala and hippocampus, particularly in humans (*Maguire et al., 2000*; *Taren et al., 2013*). Additionally, our conceptual framework opens avenues for advanced neuroimaging techniques such as diffusion tensor imaging (DTI) (*Howard et al., 2023*; *Zhang et al., 2013*) or multiparametric MRI (*Mulholland et al., 2024*), which could be used to explore white matter connectivity or microstructural changes. Our findings also emphasize the need to develop individual-level measures of social tolerance (*Dubuc et al., 2012*; *DeTroy et al., 2022*). Fine-tuning these measures would allow more precise correlations between behavioral data and neuroanatomical features. By operationalizing the concept of social tolerance on cognitive dimensions, our work aims at enriching the framework through which primate sociality is currently studied.

## Conclusion

Our study provides novel insights into the relationship between amygdala volume and social tolerance in macaques, offering an innovative perspective on the neuroanatomical basis of social cognition. Using a comparative approach across 12 macaque species, we uncovered a revealing relationship: low-tolerance species start their life with a smaller amygdala compared to their socially tolerant counterparts. In addition, intolerant species show an increase in amygdala volume, whereas highly tolerant species show the opposite trend. These findings refine conventional views of the amygdala by highlighting its broader role in both emotional regulation and complex social cognition. The observed differences in amygdala volume with respect to social tolerance grades suggest that the development and plasticity of the amygdala seem to be intricately linked to the social environment and experiences of the species. Larger amygdala in socially tolerant species may reflect an enhanced capacity to process complex social information, facilitating better social interactions, cooperative behavior, and conflict management. Alternatively, the observed increase in amygdala volume in socially intolerant species over time may be explained by heightened socio-cognitive demands, rather than being solely attributed to chronic stress or emotional reactivity. While earlier studies emphasized the role of the amygdala in stress response, recent findings are in line with our results, which suggest that amygdala functions extend to broader aspects of social cognition. These findings have profound implications for our understanding of social brain evolution as well as underscoring the importance of developmental stage and the social environment being crucial drivers of neuroanatomical adaptations. In addition, although hippocampal volume showed less pronounced and more variable differences across social grades, a credible effect was observed between 13 and 18 years of age. Across all ages, SG4 individuals consistently exhibited larger hippocampal volumes than SG1, supporting the possibility that this region also contributes to social cognitive processes in tolerant species—especially during developmental phases associated with social maturation. This study, at the interface of primatology and cognitive neuroscience, also provides a framework for investigating the impact of the social environment on brain development and paves the way for future research to unravel the complexities of brain evolution and sociality.

# Materials and methods
## Brain specimen collection

To allow comprehensive cross-species comparisons in the *Macaca* genus, a dataset of 42 *post-mortem* specimens has been constituted through collaborations with multiple research centers, each contributing unique expertise and resources (*Supplementary file 1*). The collaborating institutions included:

### The Centre de Primatologie de l'Université de Strasbourg (CdP)

Provided valuable brain data derived from 20 brain samples. Among those, one sample (*M. nigra*) was obtained as part of a collaboration with the zoo of Mulhouse (https://www.zoo-mulhouse.com/).

## Samples from INDI-PRIME-DE (*Milham et al., 2018*)

**The Japan Monkey Center** provided 5 *post-mortem* MRI acquisitions to the dataset (*Sakai et al., 2018*). **Utrecht University:** contributed to the dataset with 13 *post-mortem* MRI acquisitions (*Navarrete et al., 2018*).

## INSERM-Oxford University

5 *post-mortem* MRI acquisitions came from this collaboration. This addition offered more variety of acquired data mostly in age and sex (*Milham et al., 2018*).

## Ethical considerations

The study was conducted in accordance with ethical guidelines and was approved by the ethical committee of the Centre de Primatologie de l'Université de Strasbourg which is authorized to house NHP (registration B6732636). The research further complied with the EU Directive 2010/63/EU for animal experiments. All subjects from the CdP died of natural or accidental causes; no macaque was euthanized in the sole frame of the project. These specimens originated from CdP, and their collection followed rigorous ethical considerations. The specimens were either obtained from previous collections—where full bodies were preserved in dedicated freezers—or from individuals of the CdP that had died from natural causes. The *post-mortem* MRI data from INSERM- Oxford University were acquired from deceased animals that died of causes unrelated to the present research project. As such, the research did not require a Home Office License as defined by the Animals (Scientific Procedures) Act 1986 of the United Kingdom.

## Brain extraction technique

*Post-mortem* MRI images acquisition of macaque brains is central to our study, more specifically, in translational studies of homologous brain regions. Brain extraction is a crucial process in neuroscience research for studying the internal brain structure of animals. Through the acquisition at the CdP of 20 *post-mortem* anatomical MRI scans of brains from six different species of macaques, we were able to refine a brain extraction technique - whether previously frozen or fresh - to minimize specimen handling artefacts and obtain image quality suitable for optimal use by the scientific community. The detailed extraction technique protocol established and used for our brain extractions is available as **an appendix**. Briefly, the head is reclined forward to expose the neck, muscles are removed to access the atlanto-occipital junction, which is then incised to allow head dislocation (*Figure 1—figure supplement 2*). An osteotome and hammer are used, ensuring no cerebellar herniation. The skull cap is carefully drilled using a rotary tool and removed (see *Figure 1—figure supplement 2A and B* and *Supplementary file 2* for required tools), and the brain is extracted by severing the olfactory peduncles, internal carotid arteries, and cranial nerves. Specimens are then fixed in 10% buffered formaldehyde for 7 days (see *Figure 1—figure supplement 2D*) and in phosphate buffered saline (PBS) for 3–4 days before being placed in Fluorinert for MRI acquisition, ensuring minimal air bubbles and optimal image quality (*Sébille et al., 2019*; see *Figure 1—figure supplement 3*).

## Sampling methods and measurements

Structural images were collected through both the open access databases and collaborations (*Milham et al., 2018*; *Navarrete et al., 2018*; *Sakai et al., 2018*), but also carried out at the IRIS platform of the ICube laboratory in Strasbourg for *post-mortem* samples kept at the CdP (see *Supplementary file 3* for the information relating to the acquisition of anatomical MRI images). The final dataset consists of 42 anatomical scans after pruning data with missing age or sex information (10 individuals), with both $T_1$ and $T_2$-weighted images. Due to their different origins, the images in the dataset did not follow the exact same acquisition protocols (different scanners and acquisition parameters, *Supplementary file 3*). In addition, *post-mortem* brain preservation and perfusion protocols are different, which may also influence the images obtained. Volume measurements were performed using a semi-automatic method to register individual images to the Subcortical Atlas of the Rhesus Macaque (SARM; *Hartig et al., 2021*; *Figure 1—figure supplement 2E*). Due to the large orientation discrepancies across the research centers, the images were first manually realigned (translation and rotation) with the atlas using ITK-SNAP (*Yushkevich et al., 2006*), then non-linearly registered using ANTs (*Avants et al., 2011*). The segmentation maps of the atlas were then transported to the subject space to extract the

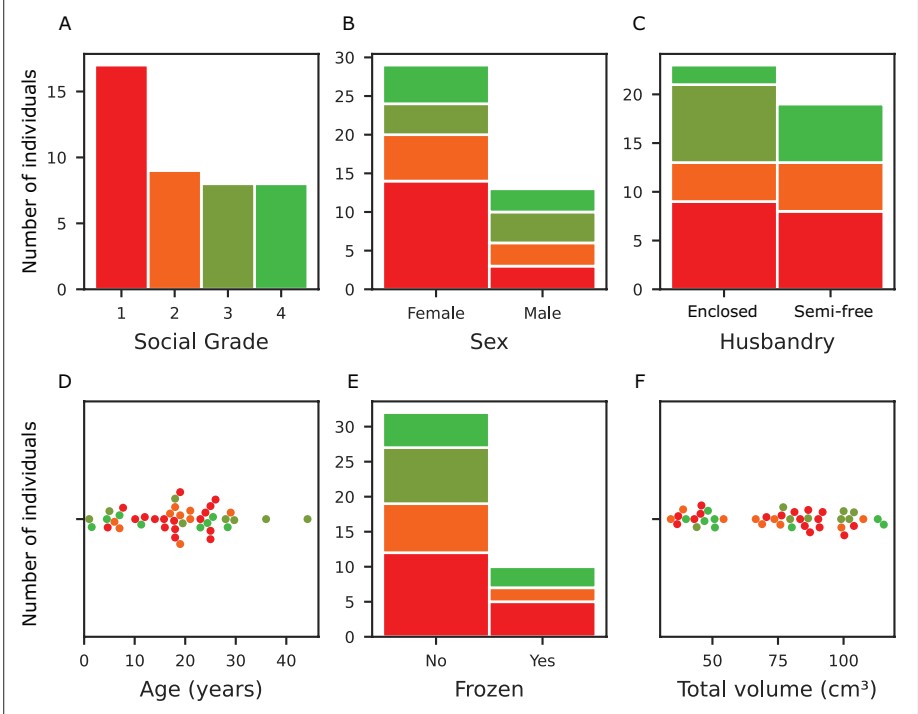

**Figure 5.** Dataset characteristics relative to the social grade. In red: social tolerance grade 1, orange: grade 2, olive: grade 3, and green: grade 4. (**A**) Social tolerance grade distribution, where grade 1 is overrepresented due to the prevalence of *Macaca mulatta* in laboratories. (**B**) Sex distribution: There was a significant imbalance in the sample, with females outnumbering males (2:1 ratio). (**C**) Husbandry distribution of the individuals (enclosed and semi-free ranging conditions) (**D**) Age distribution: The cohort had a relatively even age distribution with a notable peak in the 20 s. (**E**) The frozen status distribution. (**F**) Total brain volume distribution, excluding the myelencephalon and cerebellum due to variation in their preservation.

volume of the regions of interest. *Figure 1—figure supplement 2* details the image processing for volume extractions (see *Figure 1—figure supplement 3*). To ensure the accuracy of the SARM on our dataset, which includes 11 species other than *M. mulatta* (the species used for SARM development *Hartig et al., 2021*), we calculated the Dice Similarity Coefficient (DSC; *Zou et al., 2004*). This was done by manually segmenting, using a tablet (Wacom Cintiq 16 and ITK SNAP software), the amygdala in each acquisition and comparing the overlapping voxels between the manual segmentation and the SARM segmentation. With a DSC of 0.96, we confirm the robust performance of the SARM across our entire dataset.

## Final dataset characteristics

The dataset is composed of 12 distinct macaque species with a total of 42 individual specimens for analysis. There is a strong sex imbalance with more females than males. The age range spans from 1 to 44.20 years, with an average age of 18.2±9.4 years (standard deviation) (*Figure 5B and D*) with two outliers above 35 years old. Most importantly, based on our research question, the social grade distribution of our dataset (*Figure 5A*) is more represented by grade 1 than grade 4 species, as these species are very rare in zoos or in research centers, and most of them are protected as endangered species. The MRI acquisitions from the 42 individuals were standardized to the NMT template (*Jung et al., 2021*). Data included amygdala or hippocampus volume and a computed brain 'total volume' which only excludes the myelencephalon and the cerebellum for reliability. In fact, the integrity of these subcortical structures heavily depends on the quality and techniques used for brain extraction methods.

## Modeling approach

To investigate the subcortical correlates of social tolerance in macaques, we used a multivariate Bayesian linear model with normal likelihood, the observed data being the amygdala and hippocampus

volume. The predictors in our model were the intercept, social grade, age, sex, husbandry, whether the brain had been previously frozen, total volume, the interaction between social grade and age, and the covariance between the observations. We used wide priors, whose locations and scales were derived from the data. We assessed the quality of the model by comparing the predicted data to the observed data, and by checking the $R^2$ of the model. New data was predicted to study the interaction and the age-social grade trajectory, and the difference in volume between social grades. The predictions were made using the model on all social grades, on females aged from 1 to 40, with a total volume of 85 cm$^3$.

## Acknowledgements

The authors are grateful to the University of Strasbourg, the CNRS and Silabe (https://silabe.com/) for supporting this research and providing expert animal care. This work was further funded by ANR-21-CE37-0016 to SB and JS. This work is co-funded by the French State Region contract CPER I2MT (2014-2021), R-IRM (2021-2027) and by the European Union through the European Regional Development Fund "FEDER Grand Est". This work was performed on the IRIS platform of ICube lab, member of France Life Imaging network (grant ANR 11 INBS 0006). We extend our gratitude to the PRIME-DE open science initiative, particularly the Utrecht database, as well as the Japan Monkey Center for providing access to their open science resources. Warm thanks are extended to Brice Lefaux and his staff at Mulhouse Zoo for the collection of brain samples being made possible. Additionally, we sincerely thank Aurore de Cauwer (from ICube) for her invaluable assistance at the early stages in the MRI data acquisition. The Wellcome Centre for Integrative Neuroimaging is supported by core funding from the Wellcome Trust (203139/Z/16/Z).

## Additional information

### Funding

| Funder | Grant reference number | Author |
|---|---|---|
| Agence Nationale de la Recherche | ANR-21-CE37-0016 | Jerome Sallet<br>Sebastien Ballesta |
| Wellcome Trust | 10.35802/203139 | Jerome Sallet |
| French State Region | CPER I2MT 2014-2021 | Julien Lamy<br>Chrystelle Po |
| French State Region | R-IRM (2021-2027) | Julien Lamy<br>Chrystelle Po |
| European Union | European Regional Development Fund "FEDER Grand Est" | Julien Lamy<br>Chrystelle Po |

The funders had no role in study design, data collection and interpretation, or the decision to submit the work for publication. For the purpose of Open Access, the authors have applied a CC BY public copyright license to any Author Accepted Manuscript version arising from this submission.

### Author contributions

Sarah Silvere, Conceptualization, Data curation, Investigation, Methodology, Writing – original draft, Writing – review and editing; Julien Lamy, Data curation, Software, Formal analysis, Methodology, Writing – review and editing; Chrystelle Po, Data curation, Investigation, Methodology, Writing – review and editing; Mathieu Legrand, Investigation, Writing – review and editing; Jerome Sallet, Conceptualization, Resources, Funding acquisition, Methodology, Writing – review and editing; Sebastien Ballesta, Conceptualization, Resources, Data curation, Supervision, Funding acquisition, Investigation, Methodology, Writing – original draft, Project administration, Writing – review and editing

### Author ORCIDs

Sarah Silvere ⓘ https://orcid.org/0009-0001-4024-1453

Julien Lamy [ID] https://orcid.org/0000-0002-8400-1400
Chrystelle Po [ID] https://orcid.org/0000-0001-9785-9572
Jerome Sallet [ID] https://orcid.org/0000-0002-7878-0209
Sebastien Ballesta [ID] https://orcid.org/0000-0002-7854-5735

### Ethics

The study was conducted in accordance with ethical guidelines and was approved by the ethical committee of the Centre de Primatologie de l'Université; de Strasbourg which is authorized to house NHP (registration B6732636). The research further complied with the EU Directive 2010/63/EU for animal experiments. All subjects from the CdP died of natural or accidental causes; no macaque was euthanized in the sole frame of the project. These specimens originated from CdP, and their collection followed rigorous ethical considerations. The specimens were either obtained from previous collections-where full bodies were preserved in dedicated freezers-or from individuals of the CdP that had died from natural causes. The post-mortem MRI data from INSERM-Oxford University were acquired from deceased animals that died of causes unrelated to the present research project. As such, the research did not require a Home Office License as defined by the Animals (Scientific Procedures) Act 1986 of the United Kingdom.

Reviewer #1 (Public review): https://doi.org/10.7554/eLife.106424.3.sa1
Reviewer #2 (Public review): https://doi.org/10.7554/eLife.106424.3.sa2
Reviewer #3 (Public review): https://doi.org/10.7554/eLife.106424.3.sa3
Author response https://doi.org/10.7554/eLife.106424.3.sa4

## Additional files

### Supplementary files

Supplementary file 1. Species and data collection centers in the dataset.

Supplementary file 2. Detailed brain extraction procedure.

Supplementary file 3. Information relating to the acquisition of anatomical MRI images and the procedures for fixing and preserving *post-mortem* samples according to the different institutes.

MDAR checklist

### Data availability

The data associated with this study are available at: https://doi.org/10.17605/OSF.IO/AQMSW.

The following dataset was generated:

| Author(s) | Year | Dataset title | Dataset URL | Database and Identifier |
|---|---|---|---|---|
| Lamy J | 2025 | Neuroanatomical Foundations of Macaques' Social Tolerance: Insights from Subcortical Structures | https://doi.org/10.17605/OSF.IO/AQMSW | Open Science Framework, 10.17605/OSF.IO/AQMSW |

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

# Appendix 1

## Detailed brain extraction procedure

The first part of the acquisition of brain MRI images relies on the extraction of brain specimens. All the required materials are listed in *Supplementary file 1* and have been curated through literature recommendations (*Brown et al., 2009*; *Davenport et al., 2014*; *King et al., 2013*; *Shatil et al., 2016*) as well as through the realization of the procedure itself.

As it was the first time at the CdP that brain specimens were extracted, the development of a refined brain extraction technique played a central role in optimizing the quality of the acquired brain images. This technique, based on the literature, was meticulously applied to 17 *post-mortem* anatomical and DTI (if the specimen was not frozen) MRI scans. We optimally prepared through familiarizing ourselves with the technique and handling instruments such as the oscillating saw with a first brain extraction of a poor conditioned brain specimen (bad freezing condition). Through this first trial, we were able to adjust the following procedure as well as the chosen cutting landmarks. The brain extraction sequence consists of the following steps:

### Head dislocation

The animal is in supine position, with its head pulled dorsally to extend the neck (*Brown et al., 2009*). Then, the muscle mass of the neck has to be removed to expose the atlantooccipital junction. The junction is then incised, employing a back-and-forth motion to enable the passage of a knife or hammer axis between the cartilage (*Brown et al., 2009*).

### Adjustment for frozen specimens

For frozen specimens, a distinct approach through the atlantooccipital junction must be adopted. The section is performed with an osteotome and hammer (*Figure 1—figure supplement 1A*). Subsequently, confirmation of the absence of cerebellar protrusion through the foramen magnum is required and essential for ensuring the integrity of the specimens. If not, it would indicate cerebellar herniation and the presence of associated lesions in this brain structure (*Davenport et al., 2014*).

The head circumference was measured using a measuring tape during data collection. The average circumference of a macaque brain in our dataset is 30.5±8.7 cm. We determined the amount of formaldehyde required based on the average volume of a macaque brain. This average volume is 89.2±1.9 (SEM) for male individuals and 70.8±0.72 cm$^3$ for female individuals (*Franklin et al., 2000*; *Scott et al., 2016*). The amount of 4% formaldehyde buffered solution (pH = 6.9, Sigma-Aldrich) required to fix the brain should follow the volume ratio of tissue to be fixed to formaldehyde of 1:10 (*Thavarajah et al., 2012*), that is, a minimum of one liter of formaldehyde.

### Skull cap removal

The removal of the skull cap is the most meticulous part of the procedure aimed at providing access to the brain. This process starts with a longitudinal incision of the scalp from the anterior fontanel cranial suture to the foramen magnum. A second incision is made perpendicular to the first, along the coronal suture. The exposed skull surface underwent thorough cleaning with 70% ethanol, followed by drying with gauze pads. The bony skull cap was then delicately excised using a rotary tool (*Figure 1—figure supplement 1B*).

### Brain extraction

The extraction of the brain from the skull was conducted with care, especially for fresh specimens, as the tissues are very soft and breakable. Removal of the dura mater, cerebellum tent, and false brain is performed using a bony cap (*Brown et al., 2009*). Afterwards, the head is positioned vertically and some gentle taps on the table facilitate the gradual detachment of the brain from the skull. Employing a fluted probe, the olfactory peduncles, internal carotid arteries, and cranial nerves are delicately severed (*King et al., 2013*). The pineal gland is commonly found as a single white firm conical mass suspended at the midline of the skull cap and attached to the meninges (*King et al., 2013*). In addition, in many animals, the choroid plexus is visible as two reddish masses in a position similar to or slightly in front of the pineal gland (*King et al., 2013*). Once the brain is fully extracted, it is placed

in one of the hemispheric cups. Frozen specimens are thawed in water at room temperature before placing them in the cup (frozen tissue combined with formaldehyde fixation creates an aqueous insulating layer, altering fixation of the internal brain structures) (*Figure 1—figure supplement 1D*).

## Fixation and brain preparation for MRI

The fixation and preparation of brain specimens for MRI acquisitions is a crucial phase to ensure high quality images and low prevalence of artefacts. The chosen fixative was a 10% formaldehyde buffered solution. Ensuring complete immersion of the brain in the fixative is crucial. To achieve this, a carefully calculated volume of the solution is added to a labeled container (usually around 3 L). The brain is then placed within the container, submerged to guarantee full coverage. Sealing the container prevents contamination. The container remains for 7 days under the fume hood, and we check and gently stir every day to ensure the good repartition of the formaldehyde on the tissue. Once the brain is fixed, it is immersed the brain in a 0.5 L jar filled with PBS for 48 hr before MRI.

### 1. MRI acquisitions

The diameter of the plastic container was chosen based on the diameter of the MRI antenna used (8.6 cm in diameter).

- The sample is placed in a spherical container. The orientation in which the brain is placed (for future reference during the MRI acquisitions) was registered. Aquarium foam squares are placed around the brain to minimize the residual movements from the MRI's vibrations and to contain the remaining air bubble at the top of the container (*Figure 1—figure supplement 2B*).
- The container is filled to the brim with Fluorinert FC-770, a liquid that optimizes the contrast of the MRI signal and allows the wobble adjustment.
- The container is placed in a vacuum chamber (negative pressure of –0.1 Pa.) to limit the presence of air bubbles on the images, for up to 3 hr to remove air bubbles (*Shatil et al., 2016*; *Figure 1—figure supplement 1A*). If needed, some Fluorinert FC-770 can be added up to the brim.
- The container is sealed with parafilm (*Figure 1—figure supplement 2B*).

### 2. Recommended 7T-MRI scanning setup

Place the container with an elevating foam square to contain the remaining bubble at the top of the jar and limit the superimposition of the bubble on the brain (*Figure 1—figure supplement 2C and D*).

Once in the MRI, it is recommended to perform a sequence of localizer scans with the purpose of: (1) identifying significant distortions caused by air bubbles in the brain or MRI-compatible container, (2) accurately positioning the brain, and (3) establishing the slice positions necessary for subsequent data acquisition (*Shatil et al., 2016*).

