## [Editor Report · eLife Assessment]

This **important** work compares the size of two brain areas, the amygdala and the hippocampus, across 12 species belonging to the Macaca genus. The authors find, using a **convincing** methodological approach, that amygdala - but not hippocampal - volume varies with social tolerance grade, with high tolerance species showing larger amygdala than low tolerance species of macaques. Interestingly, their findings also suggest an inverted developmental effect, with intolerant species showing an increase in amygdala volume across the lifespan, compared to tolerant species exhibiting the opposite trend. Overall, this paper offers new insights into the neural basis of social and emotional processing.

---

## [Referee Report · Reviewer #1 (Public review)]

Summary:

This paper investigates the potential link between amygdala volume and social tolerance in multiple macaque species. Through a comparative lens, the authors considered tolerance grade, species, age, sex, and other factors that may contribute to differing brain volumes. They found that amygdala, but not hippocampal, volume differed across tolerance grades such that high-tolerance species showed larger amygdala than low-tolerance species of macaques. They also found that less tolerant species exhibited increases in amygdala volume with age, while more tolerant species showed the opposite. Given their wide range of species with varied biological and ecological factors, the authors' findings provide new, important evidence for changes in amygdala volume in relation to social tolerance grades. Contributions from these findings will greatly benefit future efforts in the field to characterize brain regions critical for social and emotional processing across species.

(1) This study demonstrates a concerted and impressive effort to comparatively examine neuroanatomical contributions to sociality in monkeys. The authors impressively collected samples from 12 macaque species with multiple datapoints across species age, sex, and ecological factors. Species from all four social tolerance grades were present. Further, the age range of the animals is noteworthy, particularly the inclusion of individuals over 20 years old.

(2) This work is the first to report neuroanatomical correlates of social tolerance grade in macaques in one coherent study. Given the prevalence of macaques as a model of social neuroscience, considerations of how socio-cognitive demands are impacted by the amygdala are highly important. The authors' findings will certainly inform future studies on this topic.

(3) The methodology and supplemental figures for acquiring brain MRI images are nicely detailed. Clear information on these parameters is crucial for future comparative interpretations of sociality and brain volume, and the authors do an excellent job of describing this process in full.

(4) The following comments were brought up during the review. In their revision, the authors have sufficiently addressed all of these comments by providing detailed responses and updating their manuscript. First, the revision clarified how much one could draw conclusions about "nature vs. nurture" from this study. Second, the revision also clarified the contributions of very young and very old animals in their correlations. Third, in their revision, the authors expanded on how their results could be interpreted in the context of multiple behavioral traits by Thierry (2021) by providing more detailed descriptions. Finally, during the revision, the authors clarified that both intolerant and tolerant species experience complex socio-cognitive demands and highlighted that socio-cognitive challenges arise across the tolerance spectrum under different behavioral demands.

---

## [Referee Report · Reviewer #2 (Public review)]

Summary:

This comparative study of macaque species and type of social interaction is both ambitious and inevitably comes with a lot of caveats. The overall conclusion is that more intolerant species have a larger amygdala. There are also opposing development profiles regarding amygdala volume depending on whether it is a tolerant or intolerant species.

To achieve any sort of power they have combined data from 4 centres - that have all used different scanning methods and there are some resolution differences. The authors have also had to group species into 4 classifications - again to assist with any generalisations and power. They have focussed on the volumes of two structures, the amygdala and the hippocampus, which seems appropriate. Neither structure is homogeneous and so it may well be that a targeted focus on specific nuclei or subfields would help (the authors may well do this next) - but as the variables would only increase further along with the number of potential comparisons, alongside small group numbers, it seems only prudent to treat these findings are preliminary. That said, it is highly unlikely that large numbers of macaque brains will become available in the near future.

This introduction is by way of saying that the study achieves what it sets out to do, but there are many reasons to see this study as preliminary. The main message seems to be twofold: (1) that more intolerant species have relatively larger amygdalae, and (2) that with development there is an opposite pattern of volume change (increasing with age in intolerant sp and decreasing with age in tolerant species). Finding 1 is the opposite of that predicted in Table 1 - this is fine, but it should be made clearer in the Discussion that this is the case otherwise the reader may feel confused. As I read it, the authors have switched their prediction in the Discussion, which feels uncomfortable.

It is inevitable that the data in a study of this complexity are all too prone to post hoc considerations, to which the authors indulge. I suspect I would end up doing the same but it feels a bit like 'heads I win, tails you lose'. In the case of Grade 1 species, the individuals have a lot to learn especially if they are not top of the hierarchy, but at the same time there are fewer individuals in the troop, making predictions very tricky. As noted above, I am concerned by the seemingly opposite predictions in Table 1 and those in the Discussion regarding tolerance and amygdala volume. (It may be that the predictions in Table 1 are the opposite to how I read them, in which case the Table and preceding text needs to align.)

Comments on revisions:

I am happy with all of the revisions and the care shown by the authors.

---

## [Referee Report · Reviewer #3 (Public review)]

Summary:

In this study, the authors were looking at neurocorrelates of behavioural differences within the genus Macaca. To do so, they engaged in real-world dissection of dead animals (unconnected to the present study) coming from a range of different institutions. They subsequently compare different brain areas, here the amygdala and the hippocampus, across species. Crucially, these species have been sorted according to different levels of social tolerance grades (from 1 to 4). 12 species are represented across 42 individuals. The sampling process has weaknesses ("only half" of the species contained by the genus, and *Macaca mulatta*, the rhesus macaque, representing 13 of the total number of individuals), but also strengths (the species are decently well represented across the 4 grades) for the given purpose and for the amount of work required here. I will not judge the dissection process as I am not a neuroanatomist, and I will assume that the different interventions do not alter volume in any significant ways / or that the different conditions in which the bodies were kept led to the documented differences across species.

There are two main results of the study. First, in line with their predictions, the authors find that more tolerant macaque species have larger amygdala, compared to the hippocampus that remains undifferentiated across species. Second, they also identify developmental effects, although with different trends: in tolerant species, the amygdala relative volume decreases across the lifespan, while in intolerant species, the contrary occurs. The modifications brought up between the two versions of the article have answered my remarks regarding age/grade/brain area differences.

As such, I think the results are holding strong, but maybe more work is needed with respect to interpretation.

Classification of the social grade, as well as the issue of nature vs nurture have been addressed by the authors, I thank them for this.

I still feel the integration of the amygdala as a common cognitive & emotional center could be possibly more pushed in the discussion, although I acknowledge that it would be complicated to do without knowing how the emotional and social lives of these animals impacted the growth of their amygdala...

Strengths:

Methods & breadth of species tested

Weaknesses:

Interpretations, which, although softened, could still be more integrated with the literature on emotion

---

## [Author Response]

The following is the authors’ response to the original reviews.

**Public Reviews:**

**Reviewer #1 (Public review):**

We thank Reviewer #1 for its thoughtful and constructive feedback. We found the suggestions particularly helpful in refining the conceptual framework and clarifying key aspects of our interpretations.

Summary:This paper investigates the potential link between amygdala volume and social tolerance in multiple macaque species. Through a comparative lens, the authors considered tolerance grade, species, age, sex, and other factors that may contribute to differing brain volumes. They found that amygdala, but not hippocampal, volume differed across tolerance grades, such that hightolerance species showed larger amygdala than low-tolerance species of macaques. They also found that less tolerant species exhibited increases in amygdala volume with age, while more tolerant species showed the opposite. Given their wide range of species with varied biological and ecological factors, the authors' findings provide new evidence for changes in amygdala volume in relation to social tolerance grades. Contributions from these findings will greatly benefit future efforts in the field to characterize brain regions critical for social and emotional processing across species.Strengths:(1) This study demonstrates a concerted and impressive effort to comparatively examine neuroanatomical contributions to sociality in monkeys. The authors impressively collected samples from 12 macaque species with multiple datapoints across species age, sex, and ecological factors. Species from all four social tolerance grades were present. Further, the age range of the animals is noteworthy, particularly the inclusion of individuals over 20 years old - an age that is rare in the wild but more common in captive settings.(2) This work is the first to report neuroanatomical correlates of social tolerance grade in macaques in one coherent study. Given the prevalence of macaques as a model of social neuroscience, considerations of how socio-cognitive demands are impacted by the amygdala are highly important. The authors' findings will certainly inform future studies on this topic.(3) The methodology and supplemental figures for acquiring brain MRI images are well detailed. Clear information on these parameters is crucial for future comparative interpretations of sociality and brain volume, and the authors do an excellent job of describing this process in full.Weaknesses:(1) The nature vs. nurture distinction is an important one, but it may be difficult to draw conclusions about "nature" in this case, given that only two data points (from grades 3 and 4) come from animals under one year of age (Method Figure 1D). Most brains were collected after substantial social exposure-typically post age 1 or 1.5-so the data may better reflect developmental changes due to early life experience rather than innate wiring. It might be helpful to frame the findings more clearly in terms of how early experiences shape development over time, rather than as a nature vs. nurture dichotomy.

We agree with the reviewer that presenting our findings through a strict nature vs. nurture dichotomy was potentially misleading. We have revised the introduction and the discussion (e.g. lines 85-95 and 363-365) to clarify that we examined how neurodevelopmental trajectories differ across social grades with the caveat of related to the absence of very young individuals in our samples. We now explicitly mention that our results may reflect both early species-typical biases and experience-dependent maturation.

We positioned our study on social tolerance in a comparative neuroscience framework and introduced a tentative working model that articulates behavioral traits, cognitive dimensions, and their potential subcortical neural substrates

Drawing upon 18 behavioral traits identified in Thierry’s comparative analyses (Thierry, 2021, 2007), we organize these traits into three core dimensions: socio-cognitive demands, behavioral inhibition, and the predictability of the social environment (Table 1). This conceptualization does not aim to redefine social tolerance itself, but rather to provide a structured basis for testing neuroanatomical hypotheses related to social style variability. It echoes recent efforts to bridge behavioral ecology and cognitive neuroscience by linking specific mental abilities – such as executive functions or metacognition – with distinct prefrontal regions shaped by social and ecological pressures (Bouret et al., 2024).

“Cross-fostering experiments (De Waal and Johanowicz, 1993), along with our own results, suggest that social tolerance grades reflect both early, possibly innate predispositions and later environmental shaping”.

(2) It would be valuable to clarify how the older individuals, especially those 20+ years old, may have influenced the observed age-related correlations (e.g., positive in grades 1-2, negative in grades 3-4). Since primates show well-documented signs of aging, some discussion of the potential contribution of advanced age to the results could strengthen the interpretation.

We thank the reviewer for highlighting this important point. In our dataset, younger and older subjects are underrepresented, but they are distributed across all subgroups. Therefore, we do not think that it could drive the interaction effect we are reporting. In our sample, amygdala volume tended to increase with age in intolerant species and decrease in tolerant species. We included a new analysis (Figure 4) that allows providing a clearer assessment of when social grades 1 vs 4 differed in terms of amygdala and hippocampus volume. While our model accounts for age continuously, we agree that age-related variation deserves cautious interpretation and require longitudinal designs in future studies.

We also added the following statements in the discussion (lines 386-391)

“Due to a limited sample size of our study, this crossing trend, already accounted for by our continuous age model, should be further investigated. These results call for cautious interpretation of age-related variation and further emphasize the importance of longitudinal studies integrating both behavioral, cognitive and anatomical data in non-human primates, which would help to better understand the link between social environment and brain development (Song et al., 2021)”.

(3) The authors categorize the behavioral traits previously described in Thierry (2021) into 3 selfdefined cognitive requirements, however, they do not discuss under what conditions specific traits were assigned to categories or justify why these cognitive requirements were chosen. It is not fully clear from Thierry (2021) alone how each trait would align with the authors' categories. Given that these traits/categories are drawn on for their neuroanatomical hypotheses, it is important that the authors clarify this. It would be helpful to include a table with all behavioral traits with their respective categories, and explain their reasoning for selecting each cognitive requirement category.

Thank you for this important suggestion. We have extensively revised the introduction to explain how we derived from the scientific literature the three cognitive dimensions—socio-cognitive demands, behavioral inhibition, and predictability of the social environment—. We now provide a complete overview of the 18 behavioral traits described in Thierry’s framework and their cognitive classification in a dedicated table , along with hypothesized neural correlates. We have also mentioned traits that were not classified in our framework along with short justification of this classification. We believe this addition significantly improves the transparency and intelligibility of our conceptual approach.

“The concept of social tolerance, central to this comparative approach, has sometimes been used in a vague or unidimensional way. As Bernard Thierry (2021) pointed out, the notion was initially constructed around variations in agonistic relationships – dominance, aggressiveness, appeasement or reconciliation behaviors – before being expanded to include affiliative behaviors, allomaternal care or male–male interactions (Thierry, 2021). These traits do not necessarily align along a single hierarchical axis but rather reflect a multidimensional complexity of social style, in which each trait may have co-evolved with others (Thierry, 2021, 2000; Thierry et al., 2004). Moreover, the lack of a standardized scientific definition has sometimes led to labeling species as “tolerant” or “intolerant” without explicit criteria (Gumert and Ho, 2008; Patzelt et al., 2014). These behavioral differences are characterized by different styles of dominance (Balasubramaniam et al., 2012), severity of agonistic interactions (Duboscq et al., 2014), nepotism (Berman and Thierry, 2010; Duboscq et al., 2013; Sueur et al., 2011) and submission signals (De Waal and Luttrell, 1985; Rincon et al., 2023), among the 18 covariant behavioral traits described in Thierry's classification of social tolerance (Thierry, 2021, 2017, 2000)”.

“To ground the investigation of social tolerance in a comparative neuroanatomical framework, we introduce a tentative working model that articulates behavioral traits, cognitive dimensions, and their potential subcortical neural substrates. Drawing upon 18 behavioral traits identified in Thierry’s comparative analyses (Thierry, 2021, 2007), we organized these traits into three core dimensions: socio-cognitive demands, behavioral inhibition, and the predictability of the social environment (Table 1). This conceptualization does not aim to redefine social tolerance itself, but rather to provide a structured basis for testing neuroanatomical hypotheses related to social style variability. It echoes recent efforts to bridge behavioral ecology and cognitive neuroscience by linking specific mental abilities – such as executive functions or metacognition – with distinct prefrontal regions shaped by social and ecological pressures (Bouret et al., 2024; Testard 2022)”.

(4) One of the main distinctions the authors make between high social tolerance species and low tolerance species is the level of complex socio-cognitive demands, with more tolerant species experiencing the highest demands. However, socio-cognitive demands can also be very complex for less tolerant species because they need to strategically balance behaviors in the presence of others. The relationships between socio-cognitive demands and social tolerance grades should be viewed in a more nuanced and context-specific manner.

We fully agree and we did not mean that intolerant species lives in a ‘simple’ social environment but that the ones of more tolerant species is markedly more demanding. Evidence supporting this statement include their more efficient social networks (Sueur et al., 2011) and more complex communicative skills (e.g. tolerant macaques displayed higher levels of vocal diversity and flexibility than intolerant macaques in social situation with high uncertainty Rebout et al., 2020).

In the revised version (lines 106-122), we now highlight that socio-cognitive challenges arise across the tolerance spectrum, including in less tolerant species where strategic navigation of rigid hierarchies and risk-prone interactions is required. We hope that this addition offers a more balanced and nuanced framing of socio-cognitive demands across macaque societies

“The first category, socio-cognitive demands, refers to the cognitive resources needed to process, monitor, and flexibly adapt to complex social environments. Linking those parameters to neurological data is at the core of the social brain theory to explain the expansion of the neocortex in primates (Dunbar). Macaques social systems require advanced abilities in social memory, perspective-taking, and partner evaluation (Freeberg et al., 2012). This is particularly true in tolerant species, where the increased frequency and diversity of interactions may amplify the demands on cognitive tracking and flexibility. Tolerant macaque species typically live in larger groups with high interaction frequencies, low nepotism, and a wider range of affiliative and cooperative behaviors, including reconciliation, coalition-building, and signal flexibility (REF). Tolerant macaque species also exhibit a more diverse and flexible vocal and facial repertoire than intolerants ones which may help reduce ambiguity and facilitate coordination in dense social networks (Rincon et al., 2023; Scopa and Palagi, 2016; Rebout 2020). Experimental studies further show that macaques can use facial expressions to anticipate the likely outcomes of social interactions, suggesting a predictive function of facial signals in managing uncertainty (Micheletta et al., 2012; Waller et al., 2016). Even within less tolerant species, like *M. mulatta*, individual variation in facial expressivity has been linked to increased centrality in social networks and greater group cohesion, pointing to the adaptive value of expressive signaling across social styles (Whitehouse et al., 2024)”.

(5) While the limitations section touches on species-related considerations, the issue of individual variability within species remains important. Given that amygdala volume can be influenced by factors such as social rank and broader life experience, it might be useful to further emphasize that these factors could introduce meaningful variation across individuals. This doesn't detract from the current findings but highlights the importance of considering life history and context when interpreting subcortical volumes-particularly in future studies.

We have now emphasized this point in the limitations section (lines 441-456). While our current dataset does not allow us to fully control for individual-level variables across all collection centers, we recognize that factors such as rank, social exposure, and individual life history may influence subcortical volumes

“Although we explained some interspecies variability, adding subjects to our database will increase statistical power and will help addressing potential confounding factors such as age or sex in future studies. One will benefit from additional information about each subject. While considered in our modelling, the social living and husbandry conditions of the individuals in our dataset remain poorly documented. The living environment has been considered, and the size of social groups for certain individuals, particularly for individuals from the CdP, have been recorded. However, these social characteristics have not been determined for all individuals in the dataset. As previously stated, the social environment has a significant impact on the volumetry of certain regions. Furthermore, there is a lack of data regarding the hierarchy of the subjects under study and the stress they experience in accordance with their hierarchical rank and predictability of social outcomes position (McCowan et al., 2022)”.

**Reviewer #2 (Public review):**

We thank Reviewer #2 for its thoughtful remarks and for acknowledging the value of our comparative approach despite its inherent constraints.

Summary:This comparative study of macaque species and the type of social interaction is both ambitious and inevitably comes with a lot of caveats. The overall conclusion is that more intolerant species have a larger amygdala. There are also opposing development profiles regarding amygdala volume depending on whether it is a tolerant or intolerant species.To achieve any sort of power, they have combined data from 4 centres, which have all used different scanning methods, and there are some resolution differences. The authors have also had to group species into 4 classifications - again to assist with any generalisations and power. They have focused on the volumes of two structures, the amygdala and the hippocampus, which seems appropriate. Neither structure is homogeneous and so it may well be that a targeted focus on specific nuclei or subfields would help (the authors may well do this next) - but as the variables would only increase further along with the number of potential comparisons, alongside small group numbers, it seems only prudent to treat these findings are preliminary. That said, it is highly unlikely that large numbers of macaque brains will become available in the near future.This introduction is by way of saying that the study achieves what it sets out to do, but there are many reasons to see this study as preliminary. The main message seems to be twofold: (1) that more intolerant species have relatively larger amygdalae, and (2) that with development, there is an opposite pattern of volume change (increasing with age in intolerant species and decreasing with age in tolerant species). Finding 1 is the opposite of that predicted in Table 1 - this is fine, but it should be made clearer in the Discussion that this is the case, otherwise the reader may feel confused. As I read it, the authors have switched their prediction in the Discussion, which feels uncomfortable.

We thank the reviewer for this important observation. In the original version, Table 1 presented simplified direct predictions linking social tolerance grades to amygdala and hippocampus volumes. We recognize that this formulation may have created confusion In the revised manuscript, we have thoroughly restructured the table and its accompanying rationale. Table 1 now better reflects our conceptual framework grounded in three cognitive dimensions—sociocognitive demands, behavioral inhibition, and social predictability—each linked to behavioral traits and associated neural hypotheses based on published literature. This updated framework, detailed in lines 144-169 of the introduction, provides a more nuanced basis for interpreting our results and avoids the inconsistencies previously noted. The Discussion was also revised accordingly (lines 329-255) to clarify where our findings diverge from the original predictions and to explore alternative explanations based on social complexity. Rather than directly predicting amygdala size from social tolerance grades, we propose that variation in volume emerges from differing combinations of cognitive pressures across species.

It is inevitable that the data in a study of this complexity are all too prone to post hoc considerations, to which the authors indulge. In the case of Grade 1 species, the individuals have a lot to learn, especially if they are not top of the hierarchy, but at the same time, there are fewer individuals in the troop, making predictions very tricky. As noted above, I am concerned by the seemingly opposite predictions in Table 1 and those in the Discussion regarding tolerance and amygdala volume. (It may be that the predictions in Table 1 are the opposite of how I read them, in which case the Table and preceding text need to align.)In order to facilitate the interpretation of our Bayesian modelling, we have selected a more focused ROI in our automatic segmentation procedure of the Hippocampus (from Hippocampal Formation to Hippocampus) and have added to the new analysis (Figure 4) that helps to properly test whether the hippocampus significantly differs between species from social grade 1 vs 4. The present analysis found that this is the case in adult monkeys. This is therefore consistent with our hypothesis that amygdala volumes are principally explained by heightened sociocognitive demands in more tolerant species.

We also acknowledge the reviewer’s concerns about the limited generalizability due to our sample. The challenges of comparative neuroimaging in non-human primates—especially when using post-mortem datasets—are substantial. Given the ethical constraints and the rarity of available specimens, increasing the number of individuals or species is not feasible in the short term. However, we have made all data and code publicly available and clearly stated the limitations of our sample in the manuscript. Despite these constraints, we believe our dataset offers an unprecedented comparative perspective, particularly due to the inclusion of rare and tolerant species such as M. tonkeana, M. nigra, and M. thibetana, which have never been included in structural MRI studies before. We hope this effort will serve as a foundation for future collaborative initiatives in primate comparative neuroscience.

**Reviewer #3 (Public review):**

We thank Reviewer #3 for their thoughtful and detailed review. Their comments helped us refine both the conceptual and interpretative aspects of the manuscript. We respond point by point below.

Summary:In this study, the authors were looking at neurocorrelates of behavioural differences within the genus Macaca. To do so, they engaged in real-world dissection of dead animals (unconnected to the present study) coming from a range of different institutions. They subsequently compare different brain areas, here the amygdala and the hippocampus, across species. Crucially, these species have been sorted according to different levels of social tolerance grades (from 1 to 4). 12 species are represented across 42 individuals. The sampling process has weaknesses ("only half" of the species contained by the genus, and *Macaca mulatta*, the rhesus macaque, representing 13 of the total number of individuals), but also strengths (the species are decently well represented across the 4 grades) for the given purpose and for the amount of work required here. I will not judge the dissection process as I am not a neuroanatomist, and I will assume that the different interventions do not alter volume in any significant ways / or that the different conditions in which the bodies were kept led to the documented differences across species.25 brains were extracted by the authors themselves who are highly with this procedure. Overall, we believe that dissection protocols did not alter the total brain volume. Despite our expertise, we experienced some difficulties to not damage the cerebellum. Therefore, this region was not included in our analysis. We also noted that this brain region was also damaged or absent from the Prime-DE dataset.Several protocols were used to prepare and store tissue. It could have impacted the total brain volume.

We agree that differences in tissue preparation and storage could potentially affect total brain volume. Therefore, we explicitly included the main sample preparation variable — whether brains had been previously frozen — as a covariate in our model. This factor did not explain our results. Moreover, Figures 1D and 1I display the frozen status and its correlation with the amygdala and hippocampus ratios, respectively. Figure 2 shows the parameters of the model and the posterior distributions for the frozen status and total brain volume effects.

There are two main results of the study. First, in line with their predictions, the authors find that more tolerant macaque species have larger amygdala, compared to the hippocampus, which remains undifferentiated across species. Second, they also identify developmental effects, although with different trends: in tolerant species, the amygdala relative volume decreases across the lifespan, while in intolerant species, the contrary occurs. The results look quite strong, although the authors could bring up some more clarity in their replies regarding the data they are working with. From one figure to the other, we switch from model-calculated ratio to modelpredicted volume. Note that if one was to sample a brain at age 20 in all the grades according to the model-predicted volumes, it would not seem that the difference for amygdala would differ much across grades, mostly driven with Grade 1 being smaller (in line with the main result), but then with Grade 2 bigger than Grade 3, and then Grade 4 bigger once again, but not that different from Grade 2.Overall, despite this, I think the results are pretty strong, the correlations are not to be contested, but I also wonder about their real meaning and implications. This can be seen under 3 possible aspects:(1) Classification of the social gradeWhile it may be familiar to readers of Thierry and collaborators, or to researchers of the macaque world, there is no list included of the 18 behavioral traits used to define the three main cognitive requirements (socio-cognitive demands, predictability of the environment, inhibitory control). It would be important to know which of the different traits correspond to what, whether they overlap, and crucially, how they are realized in the 12 study species, as there could be drastic differences from one species to the next. For now, we can only see from Table S1 where the species align to, but it would be a good addition to have them individually matched to, if not the 18 behavioral traits, at least the 3 different broad categories of cognitive requirements.

We fully agree with this observation. In the revised version of the manuscript, we now include a detailed conceptual table listing all 18 behavioral traits from Thierry’s framework. For each trait, we provide its underlying social implications, its associated cognitive dimension (when applicable), and the hypothesized neural correlate.

While some traits may could have been arguably classified in several cognitive dimensions (e.g. reconciliation rate), we preferred to assign each to a unique dimension for clarity. Additionally, the introduction (lines 95-169 + Table1) now explains how each trait was evaluated based on existing literature and assigned to one of the three proposed cognitive categories: socio-cognitive demands, behavioral inhibition, or social unpredictability. This structure offers a clearer and more transparent basis for the neuroanatomical hypotheses tested in the study.

“Navigating social life in primate societies requires substantial cognitive resources: individuals must not only track multiple relationships, but also regulate their own behavior, anticipate others’ reactions, and adapt flexibly to changing social contexts. Taken advantage of databases of magnetic resonance imaging (MRI) structural scans, we conducted the first comparative study integrating neuroanatomical data and social behavioral data from closely related primate species of the same genus to address the following questions: To what extent can differences in volumes of subcortical brain structures be correlated with varying degrees of social tolerance? Additionally, we explored whether these dispositions reflect primarily innate features, shaped by evolutionary processes, or acquired through socialization within more or less tolerant social environments”.

“The first category, socio-cognitive demands, refers to the cognitive resources needed to process, monitor, and flexibly adapt to complex social environments. Linking those parameters to neurological data is at the core of the social brain theory to explain the expansion of the neocortex in primates (Dunbar). Macaques social systems require advanced abilities in social memory, perspective-taking, and partner evaluation (Freeberg et al., 2012). This is particularly true in tolerant species, where the increased frequency and diversity of interactions may amplify the demands on cognitive tracking and flexibility. Tolerant macaque species typically live in larger groups with high interaction frequencies, low nepotism, and a wider range of affiliative and cooperative behaviors, including reconciliation, coalition-building, and signal flexibility (REF). Tolerant macaque species also exhibit a more diverse and flexible vocal and facial repertoire than intolerants ones which may help reduce ambiguity and facilitate coordination in dense social networks (Rincon et al., 2023; Scopa and Palagi, 2016; Rebout 2020). Experimental studies further show that macaques can use facial expressions to anticipate the likely outcomes of social interactions, suggesting a predictive function of facial signals in managing uncertainty (Micheletta et al., 2012; Waller et al., 2016). Even within less tolerant species, like *M. mulatta*, individual variation in facial expressivity has been linked to increased centrality in social networks and greater group cohesion, pointing to the adaptive value of expressive signaling across social styles (Whitehouse et al., 2024)”.

“The second category, inhibitory control, includes traits that involve regulating impulsivity, aggression, or inappropriate responses during social interactions. Tolerant macaques have been shown to perform better in tasks requiring behavioral inhibition and also express lower aggression and emotional reactivity in both experimental and natural contexts (Joly et al., 2017; Loyant et al., 2023). These features point to stronger self-regulation capacities in species with egalitarian or less rigid hierarchies. More broadly, inhibition – especially in its strategic form (self-control) – has been proposed to play a key role in the cohesion of stable social groups. Comparative analyses across mammals suggest that this capacity has evolved primarily in anthropoid primates, where social bonds require individuals to suppress immediate impulses in favour of longer-term group stability (Dunbar and Shultz, 2025). This view echoes the conjecture of Passingham and Wise (2012), who proposed that the emergence of prefrontal area BA10 in anthropoids enabled the kind of behavioural flexibility needed to navigate complex social environments (Passingham et al., 2012)”.

“The third category, social environment predictability, reflects how structured and foreseeable social interactions are within a given society. In tolerant species, social interactions are more fluid and less kin-biased, leading to greater contextual variation and role flexibility, which likely imply a sustained level of social awareness. In fact, as suggested by recent research, such social uncertainty and prolonged incentives are reflected by stress-related physiology : tolerant macaques such as M. tonkeana display higher basal cortisol levels, which may be indicative of a chronic mobilization of attentional and regulatory resources to navigate less predictable social environments (Sadoughi et al., 2021)”.

“Each behavioral trait was individually evaluated based on existing empirical literature regarding the types of cognitive operations it likely involves. When a primary cognitive dimension could be identified, the trait was assigned accordingly. However, some behaviors – such as maternal protection, allomaternal care, or delayed male dispersal – do not map neatly onto a single cognitive process. These traits likely emerge from complex configurations of affective and socialmotivational systems, and may be better understood through frameworks such as attachment theory (Suomi, 2008), which emphasizes the integration of social bonding, emotional regulation, and contextual plasticity. While these dimensions fall beyond the scope of the present framework, they offer promising directions for future research, particularly in relation to the hypothalamic and limbic substrates of social and reproductive behavior”.

“Rather than forcing these traits into potentially misleading categories, we chose to leave them unclassified within our current cognitive framework. This decision reflects both a commitment to conceptual clarity and the recognition that some behaviors emerge from a convergence of cognitive demands that cannot be neatly isolated. This tripartite framework, leaving aside reproductive-related traits, provides a structured lens through which to link behavioral diversity to specific cognitive processes and generate neuroanatomical predictions”.

(2) Issue of nature vs nurtureAnother way to look at the debate between nature vs nurture is to look at phylogeny. For now, there is no phylogenetic tree that shows where the different grades are realized. For example, it would be illuminating to know whether more related species, independently of grades, have similar amygdala or hippocampus sizes. Then the question will go to the details, and whether the grades are realized in particular phylogenetic subdivisions. This would go in line with the general point of the authors that there could be general species differences.

As pointed out by Thierry and collaborators, the social tolerance concept is already grounded in a phylogenetic framework as social tolerance matches the phylogenetical tree of these macaque species, suggesting a biological ground of these behavioral observations. Given the modest sample size and uneven species representation, we opted not to adopt tools such as Phylogenetic Generalized Least Squares (PGLS) in our analysis. Our primary aim in this study was to explore neuroanatomical variation as a function of social traits, not to perform a phylogenetic comparative analysis per see. That said, we now explicitly acknowledge this limitation in the Discussion and indicate that future work using larger datasets and phylogenetic methods will be essential to disentangle social effects from evolutionary relatedness. We hope that making our dataset openly available will facilitate such futures analyses.

With respect to nurture, it is likely more complicated: one needs to take into account the idiosyncrasies of the life of the individual. For example, some of the cited literature in humans or macaques suggests that the bigger the social network, the bigger the brain structure considered. Right, but this finding is at the individual level with a documented life history. Do we have any of this information for any of the individuals considered (this is likely out of the scope of this paper to look at this, especially for individuals that did not originate from CdP)?

We appreciate this insightful observation. Indeed, findings from studies in humans and nonhuman primates showing associations between brain structure and social network size typically rely on detailed life history and behavioral data at the individual level. Unfortunately, such finegrained information was not consistently available across our entire sample. While some individuals from the Centre de Primatologie (CdP) were housed in known group compositions and social settings, we did not have access to longitudinal social data—such as rank, grooming rates, or network centrality—that would allow for robust individual-level analyses. We now acknowledge this limitation more clearly in the Discussion (lines 436-443), and we fully agree that future work combining neuroimaging with systematic behavioral monitoring will be necessary to explore how species-level effects interact with individual social experience.

(3) Issue of the discussion of the amygdala's functionThe entire discussion/goal of the paper, states that the amygdala is connected to social life. Yet, before being a "social center", the amygdala has been connected to the emotional life of humans and non-humans alike. The authors state L333/34 that "These findings challenge conventional expectations of the amygdala's primary involvement in emotional processes and highlight the complexity of the amygdala's role in social cognition". First, there is no dichotomy between social cognition and emotion. Emotion is part of social cognition (unless we and macaques are robots). Second, there is nowhere in the paper a demonstration that the differences highlighted here are connected to social cognition differences per se. For example, the authors have not tested, say, if grade 4 species are more afraid of snakes than grade 1 species. If so, one could predict they would also have a bigger amygdala, and they would probably also find it in the model. My point is not that the authors should try to correlate any kind of potential aspect that has been connected to the amygdala in the literature with their data (see for example the nice review by DomínguezBorràs and Vuilleumier, https://doi.org/10.1016/B978-0-12-823493-8.00015-8), but they should refrain from saying they have challenged a particular aspect if they have not even tested it. I would rather engage the authors to try and discuss the amygdala as a multipurpose center, that includes social cognition and emotion.

We thank the reviewer for this important and nuanced point. We have revised the manuscript to adopt a more cautious and integrative tone regarding the function of the amygdala. In the revised Discussion (lines 341-355), we now explicitly state that the amygdala is involved in a broad range of processes—emotional, social, and affective—and that these domains are deeply intertwined. Rather than proposing a strict dissociation, we now suggest that the amygdala supports integrated socio-emotional functions that are mobilized differently across social tolerance styles. We also cite recent relevant literature (e.g., Domínguez-Borràs & Vuilleumier, 2021) to support this view and have removed any claim suggesting we challenge the emotional function of the amygdala per se. Our aim is to contribute to a richer understanding of how affective and social processes co-construct structural variation in this region.

Strengths:Methods & breadth of species tested.Weaknesses:Interpretation, which can be described as 'oriented' and should rather offer additional views.
**Recommendations for the authors:**

**Reviewer #1 (Recommendations for the authors):**
Private Comments:(1) Table 1 should be formatted for clarity i.e., bolded table headers, text realignment, and spacing. It was not clear at first glance how information was organized. It may also be helpful to place behavioral traits as the first column, seeing that these traits feed into the author's defined cognitive requirements.

We have reformatted Table 1 to improve clarity and readability. Behavioral traits now appear in the first column, followed by cognitive dimensions and hypothesized neural correlates. Column headers have been bolded and alignment has been standardized.

(2) Figures could include more detail to help with interpretations. For example, Figure 3 should define values included on the x-axis in the figure caption, and Figure 4 should explain the use of line, light color, and dark color. Figure 1 does not have a y-axis title.

The figures have been revised and legends completed to ensure more clarity.

(3) Please proofread for typos throughout.

The manuscript has been carefully proofread, and all typographical and grammatical errors have been corrected. These changes are visible in the tracked version.

**Reviewer #2 (Recommendations for the authors):**
Specific comments:(1) Given all of the variability would it not be a good idea to just compare (eg in the supplemental) the macaque data from just the Strasbourg centre for m mulatta and m toneanna. I appreciate the ns will be lower, but other matters are more standardized.

We fully understand the reviewer’s suggestion to restrict the comparison to data collected at a single site in order to minimize inter-site variability. However, as noted, such an analysis would come at the cost of statistical power, as the number of individuals per species within a single center is small. For example, while M. tonkeana is well represented at the Strasbourg centre, only one individual of *M. mulatta* is available from the same site. Thus, a restricted comparison would severely limit the interpretability of results, particularly for age-related trajectories. To address variability, we included acquisition site and brain preservation method as covariates or predictors where appropriate, and we have been cautious in our interpretations. We also now emphasize in the Methods and Discussion the value of future datasets with more standardized acquisition protocols across species and centers. We hope that by openly sharing our data and workflow, we can contribute to this broader goal.

(2) I have various minor edits:(a) L 25 abstract - Specify what is meant by 'opposite trend'; the reader cannot infer what this is.

Modified in line 25-28: “Unexpectedly, tolerant species exhibited a decrease in relative amygdala volume across the lifespan, contrasting with the age-related increase observed in intolerant species—a developmental pattern previously undescribed in primates.”

(b) L67 - The reference 'Manyprimates' needs fixing as it does in the references section.

After double checking, Manyprimates studies are international collaborative efforts that are supposed to be cite this way (https://manyprimates.github.io/#pubs).

(c) L74 - Taking not Taken.

This typo has been corrected.

(d) L129 - It says 'total volume', but this is corrected total volume?

We have clarified in the figures legends that the “total brain volume” used in our analyses excludes the cerebellum and the myelencephalon, as specified in our image preprocessing protocol. This ensures consistency across individuals and institutions.

(e) L138 - Suddenly mentions 'frozen condition' without any prior explanation - this needs explaining in the legend - also L144.

We have added an explanation of the ‘frozen condition’ variable in in the relevant figure legend.

(f) L166 - Results - it would be helpful to remind readers what Grade 1 signifies, ie intolerant species.

We now include a brief reminder in the Results section that Grade 1 corresponds to socially intolerant species, to help readers unfamiliar with the classification (Lines 240-251).

(g)Figure 4 - Provide the ns for each of the 4 grades to help appreciate the meaningfulness of the curves, etc.The number of subjects has been added to the Figure and a novel analysis helps in the revised ms help to appreciate the meaningfulness of some of these curves.(h) L235 - 'we had assumed that species of high social tolerance grade would have presented a smaller amygdala in size compared to grade 1'. But surely this is the exact opposite of what is predicted in Table 1 - ie, the authors did not predict this as I read the paper (Unless Table l is misleading/ambiguous and needs clarification).

As discussed in our response to Reviewer #2 and #3, we have restructured both Table 1 and the Discussion to ensure consistency. We now explicitly state that the findings diverge from our initial inhibitory-control-based prediction and propose alternative interpretations based on sociocognitive demands.

(i) L270 - 'This observation' which?? Specify.

We have replaced ‘this observation’ with a precise reference to the observed developmental decrease in amygdala volume in tolerant species.

(j) L327 - 'groundbreaking' is just hype given that there are so many caveats - I personally do not like the word - novel is good enough.

We have replaced the word ‘groundbreaking’ with ‘novel’ to adopt a more measured and appropriate tone in the discussion.

(3) I might add that I am happy with the ethics regarding this study.Thanks, we are also happy that we were able to study macaque brains from different species using opportunistic samplings along with already available data. We are collectively making progress on this!(4) Finally, I should commend the authors on all the additional information that they provide re gender/age/species. Given that there are 2xs are many females as males, it would be good to know if this affects the findings. I am not a primatologist, so I don't know, for example, if the females in Grade 1 monkeys are just as intolerant as the males?

We thank the reviewer for this thoughtful comment. We now explicitly mention the female-biased sex ratio in the Methods section and report in the Results (Figure 2, Figure 3) that sex was included as a covariate in our Bayesian models. While a small effect of sex was found for hippocampal volume, no effect was observed for the amygdala. Given the strong imbalance in our dataset (2:1 female-to-male ratio), we refrained from drawing any conclusion about sex-specific patterns, as these would require larger and more balanced samples. Although we did not test for sex-by-grade interactions, we agree that this question—especially regarding whether females and males express social style differences similarly across grades—represents an important direction for future comparative work.

**Reviewer #3 (Recommendations for the authors):**
I found the article well-written, and very easy to follow, so I have little ways to propose improvements to the article to the authors, besides addressing the various major points when it comes to interpretation of the data.One list I found myself wanting was in fact the list of the social tolerance grades, and the process by which they got selected into 3 main bags of socio-cognitive skills. Then it would become interesting to see how each of the 12 species compares within both the 18 grades (maybe once again out of the scope of this paper, there are likely reviews out there that already do that, but then the authors should explicitly mention so in the paper: X, 19XX have compared 15 out of 18 traits in YY number of macaque species); and within the 3 major subcognitive requirements delineated by the authors, maybe as an annex?

We thank the reviewer for this thoughtful suggestion. In the revised manuscript, we now include a detailed table (Table 1) that lists the 18 behavioral traits derived from Thierry’s framework, along with their associated cognitive dimension and hypothesized neuroanatomical correlate. While we did not create a matrix mapping each of the 12 species across all 18 traits due to space and data availability constraints, we agree this is an important direction that should be tackled by primatologist. We now include a sentence (line 87-90) in the manuscript to guide readers to previous comparative reviews (e.g., Thierry, 2000; Thierry et al., 2004, 2021) that document the expression of these traits across macaque species. We also clarify that our three cognitive categories are conceptual tools intended to structure neuroanatomical predictions, and not formal clusters derived from quantitative analyses.

In the annex, it would also be good to have a general summarizing excel/R file for the raw data, with important information like age, sex, and the relevant calculated volumes for each individual. The folders available following the links do not make it an easy task for a reader to find the raw data in one place.

We fully agree with the reviewer on the importance of data accessibility. We have now uploaded an additional supplementary file in .csv format on our OSF repository, which includes individuallevel metadata for all 42 macaques: species, sex, age, social grade, total brain volume, amygdala volume, and hippocampus volume. The link to this file is now explicitly mentioned in the Data Availability section. We hope this will facilitate comparisons with other datasets and improve usability for the community. In addition, we provide in a supplementary table the raw data that were used for our Bayesian modelling (see below).

The availability of the raw data would also clear up one issue, which I believe results from the modelling process: it looks odd on Figure 2, that volume ratios, defined as the given brain area volume divided by the total brain volume, give values above 1 (especially for the hippocampus). As such, the authors should either modify the legend or the figure. In general, it would be nicer to have the "real values" somewhere easily accessible, so that they can be compared more broadly with: (1) other macaques species to address questions relevant to the species; (2) other primates to address other questions that are surely going to arise from this very interesting work!

We thank the reviewer for pointing this out. The ratio values in Figure 1 correspond to the proportion of the regional volume (amygdala or hippocampus) relative to the total brain volume, excluding the cerebellum and myelencephalon. As such, values above 0.01 (i.e., above 1% of the brain volume) are expected for these structures and do not indicate an error. We have updated the figure legend to clarify this point explicitly. In addition, we have now made a cleaned .csv file available via OSF, containing all raw volumetric data and metadata in a format that facilitates cross-species or cross-study comparisons. This replaces the previous folder-based structure, which may have been less accessible.

Typos:L233: delete 'in'L430: insert space in 'NMT template(Jung et al., 2021).'